# Manipulation on active electronic states of metastable phase β-NiMoO₄ for large current density hydrogen evolution

Zengyao Wang[1,2,9], Jiyi Chen [1,3,4,9], Erhong Song [5,9], Ning Wang[6], Juncai Dong [7], Xiang Zhang [8], Pulickel M. Ajayan [8], Wei Yao[1], Chenfeng Wang[1], Jianjun Liu [5✉], Jianfeng Shen [1✉] & Mingxin Ye[1✉]

Non-noble transition metal oxides are abundant in nature. However, they are widely regarded as catalytically inert for hydrogen evolution reaction (HER) due to their scarce active electronic states near the Fermi-level. How to largely improve the HER activity of these kinds of materials remains a great challenge. Herein, as a proof-of-concept, we design a non-solvent strategy to achieve phosphate substitution and the subsequent crystal phase stabilization of metastable β-NiMoO₄. Phosphate substitution is proved to be imperative for the stabilization and activation of β-NiMoO₄, which can efficiently generate the active electronic states and promote the intrinsic HER activity. As a result, phosphate substituted β-NiMoO₄ exhibits the optimal hydrogen adsorption free energy (−0.046 eV) and ultralow overpotential of −23 mV at 10 mA cm⁻² in 1 M KOH for HER. Especially, it maintains long-term stability for 200 h at the large current density of 1000 mA cm⁻² with an overpotential of only −210 mV. This work provides a route for activating transition metal oxides for HER by stabilizing the metastable phase with abundant active electronic states.

[1] Institute of Special Materials and Technology, Fudan University, Shanghai, China. [2] Department of Chemistry, Fudan University, Shanghai, China. [3] Joint School of National University of Singapore and Tianjin University, International Campus of Tianjin University, Binhai New City, Fuzhou, China. [4] Department of Chemical and Biomolecular Engineering, National University of Singapore, Southeast Asia, Singapore. [5] State Key Laboratory of High Performance Ceramics and Superfine Microstructure, Shanghai Institute of Ceramics, Chinese Academy of Sciences, Shanghai, China. [6] Institute of Environment and Life, Beijing University of Technology, Beijing, PR China. [7] Beijing Synchrotron Radiation Facility, Institute of High Energy Physics, Chinese Academy of Science, Beijing, China. [8] Department of Materials Science and Nano Engineering, Rice University, Houston, USA. [9]These authors contributed equally: Zengyao Wang, Jiyi Chen, Erhong Song. ✉email: jliu@mail.sic.ac.cn; jfshen@fudan.edu.cn; mxye@fudan.edu.cn

Hydrogen, a kind of chemical with the highest gravimetric energy density and zero $CO_2$ emission after combustion, has been deemed as a promising alternative for fossil fuels for many years[1,2]. In recent years, hydrogen production through the electrochemical route is among the hottest topics. Until now, quantities of electrocatalysts have been investigated to be available for hydrogen evolution reaction (HER), while the trade-off between activity and economic cost remains a problem for industrial application[3,4]. Non-noble transition metal oxides (TMOs) are common and cost-effective substances with abundant reserves. However, poor conductivity and low activity limit their further application as active electrocatalysts for HER[5–7]. To improve this situation, substantial efforts have been devoted. In general, oxygen vacancy ($V_o$) engineering[8,9] and non-metal doping (P, S, N et al)[10–12] are used to elevate their catalytic activity[13,14]. Virtually, these strategies excite the active electronic states near the Fermi-level ($E_F$) in the system, which are indispensable for HER[15,16]. Besides, regular nano-morphology construction[17], exposed facet control[18], and crystal phase regulation[19] are useful methods to improve TMOs' electrocatalytic activity as well.

As a typical non-noble TMO, α-nickel molybdate (α-NiMoO$_4$) has attracted great attention but performed terribly in HER due to the scarcity of active electronic states near the $E_F$[17,20]. Various efforts such as the defect engineering[4,21,22], pressure-inducing[23,24], and element doping strategies are applied to enhance its HER activity[25–27] Phase transformation is also considered to be an effective method to improve catalytic ability for HER[28]. Due to the different structure and electronic arrangement, metastable phase β-NiMoO$_4$ may generate active electronic states to elevate HER activity, while its thermodynamic instability below 200 °C is yet a great challenge for HER application[19,29]. To this end, stabilizing and activating β-NiMoO$_4$ at room temperature is critical. Considering the strong electron-donating ability of P atom, the doping of P to excite the active electronic state and stabilize the metastable structure is believed feasible[30–32].

Herein, we design a non-solvent strategy to achieve phosphate substitution and the subsequent metastable crystal phase stabilization of NiMoO$_4$, which could efficiently manipulate the active electronic states of NiMoO$_4$ and promote its intrinsic HER activity. Upon structural characterizations and analysis, it is found that the phenomena of phosphate substitution appear in β-NiMoO$_4$ due to the same tetrahedral spatial configuration, effectively improving the HER activity of β-NiMoO$_4$ system. As a result, phosphate substituted β-NiMoO$_4$ (P-β-NiMoO$_4$, P-NiMoHZ sample) with rich active electronic states exhibits a superior performance with an ultralow overpotential of −23 mV at 10 mA cm$^{-2}$ and 44 mV dec$^{-1}$ Tafel slope in 1 M KOH, which is even better than the benchmark Pt/C electrocatalyst[17]. Furthermore, it displays surprisingly excellent stability and activity at large current density (1000 mA cm$^{-2}$) for 200 h with overpotential of only −210 mV. Theoretical calculations further unveil that Ni site of P-β-NiMoO$_4$ is in favor of the water dissociation, while the O1 site connecting P atom and Ni atom had the most appropriate hydrogen binding energy (−0.046 eV) for hydrogen desorption. More importantly, the regulation of the intrinsic charge distribution of exposed atoms in P-β-NiMoO$_4$ further optimizes the HER performance.

## Results

**Synthesis and structural characterization**. As illustrated in Fig. 1a, P-NiMoHZ was synthesized through a three-step process. Nickel molybdate hydrate (NiMoO$_4$·xH$_2$O) nanorods were firstly grown on the nickel foam via a hydrothermal method[33]. During the Sublimation-Vapor Phase transformation (SVPT) process[34],

ligand powder of 2-methylimidazole (2-MIM) gradually sublimated into gas in the quartz crucible (Supplementary Fig. 1), enabling the in-situ conversion from NiMoO$_4$·xH$_2$O into core-shell NiMoO$_4$-hybrid zeolitic imidazolate framework (NiMoHZ) nanorods. Upon phosphating, the outer hybrid zeolitic imidazolate framework (HZIF) coating would turn into a nitrogen-doped porous carbon shell, which is favorable for the stabilization of the inner core metastable phase β-NiMoO$_4$. Eventually, the P-NiMoHZ (500 °C, 60 min, 90 mg) was achieved (synthetic routines of other samples were given in Supplementary Fig. 2). The morphology evolution during the SVPT process (Supplementary Fig. 3) and X-ray diffraction (XRD) patterns (Supplementary Fig. 4) demonstrate the successful preparation of NiMoHZ. Besides, further phosphating did not change the nanorod array structure (Supplementary Fig. 5). XRD patterns of P-NiMoHZ (Supplementary Fig. 6) obtained at different temperatures reveal that the inner hydrate oxide gradually lost water of crystallization and completely transformed to β-NiMoO$_4$ (JCPDS No. 45-0142), a distinct crystal phase in contrast to α-NiMoO$_4$ (JCPDS No. 33-0948)[17,33]. It should be noted that when phosphating temperature increased to 550 °C, the oxide was converted into a composite of Ni$_3$P, Ni$_3$(PO$_4$)$_2$, and MoO$_2$, while still keeping the initial nanorod structure (Supplementary Fig. 7).

The XRD patterns of NiMoO$_4$ and P-NiMoHZ are shown in Fig. 1b. All diffraction peaks of NiMoO$_4$ match well to α-NiMoO$_4$, while those of P-NiMoHZ could be indexed to β-NiMoO$_4$. Thus, it demonstrates that the process of phosphate treatment has improved the thermodynamically stability of β-NiMoO$_4$ at ambient temperature compared with the previous report[29]. Furthermore, X-ray photoelectron spectroscopy (XPS) was performed to explore the variation of the various elements' chemical valence[18,35,36]. As shown in Fig. 1c, two peaks with binding energy of 134.3 eV and 135.1 eV correspond to the $2p_{1/2}$ and $2p_{3/2}$ of P$^{5+}$, indicating the presence of the PO$_4^{3-}$ group in P-NiMoHZ[32,37]. In contrast, no phosphorus peak is observed in NiMoO$_4$ (α-NiMoO$_4$). Given that the spatial configurations of PO$_4^{3-}$ and Mo–O polyhedra in β-NiMoO$_4$ are both tetrahedrons, it is likely that PO$_4^{3-}$ could take the place of Mo–O in P-NiMoHZ during the reaction (phosphate substitution)[32]. Besides, the O 1$s$ spectrum of NiMoO$_4$ (Supplementary Fig. 8a) could be deconvoluted into two peaks at 529.8 eV and 532.6 eV, which belong to lattice oxygen and O-H from water molecules absorbed on the surface of material[17]. For P-NiMoHZ, two same peaks as the above could also be observed. However, a peak emerges at 531.2 eV, corresponding to the O atoms in the vicinity of oxygen vacancies ($V_o$)[22]. This structure was further confirmed by the electron paramagnetic resonance (EPR) spectra in Fig. 1d. A strong centrosymmetric signal peak with the g-value of 2.003 related to the existence of $V_o$ appears in P-NiMoHZ, while no peak appears in NiMoO$_4$[8]. Moreover, compared to NiMoO$_4$, the variation in Mo 3$d$ spectrum of P-NiMoHZ also results from the emerging of $V_o$ (Supplementary Fig. 8b). Two peaks at 232.7 eV and 235.9 eV are assigned to Mo$^{6+}$[38], although the values show a slightly negative shift in contrast to the deconvoluted peaks of NiMoO$_4$, suggesting that the environments around Mo atoms in P-NiMoHZ were different from those in NiMoO$_4$. More importantly, the peaks of low-valence states (Mo$^{5+}$) indicate $V_o$ merely exists in P-NiMoHZ[17], which is in line with O 1$s$ XPS and EPR spectra results.

Except for macroscopic characterizations, high-resolution high-angle annular dark-field scanning transmission electron microscopy (HAADF-STEM) was applied to observe the crystal structure of NiMoO$_4$ and P-NiMoHZ in nanoscale[39,40]. The atomic arrangement of NiMoO$_4$ matches well with the theoretical (101) plane of the α-NiMoO$_4$ unit cell (Supplementary Fig. 9).

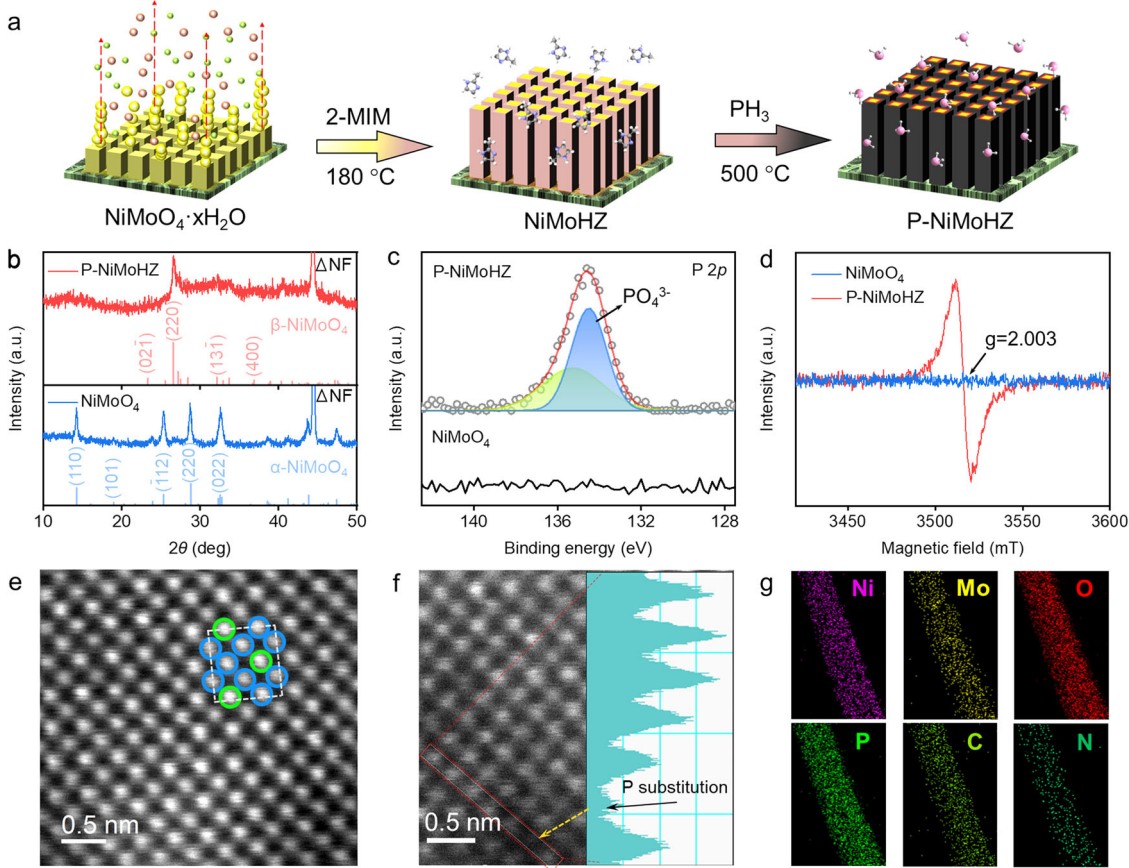

**Fig. 1 Design and structure characterization of α-NiMoO₄ and β-NiMoO₄. a** Schematic illustration of the preparation process for P-NiMoHZ. The light green, light red, and yellow balls represent Ni²⁺ cations, MoO₄²⁻ anions, and NiMoO₄ crystals, respectively. The ball-and-stick models around nanorod arrays are 2-MIM (middle) and PH₃ (right) molecules. **b** XRD patterns of P-NiMoHZ and NiMoO₄. **c** XPS spectra of P 2*p*. **d** EPR spectra of P-NiMoHZ and NiMoO₄. **e** HAADF-STEM image of P-NiMoHZ. The blue and green circles represent Mo and Ni atoms, respectively. **f** HAADF-STEM image of P-NiMoHZ with an intensity profile corresponding to the red line. **g** HAADF-STEM elemental mappings of P-NiMoHZ.

The periodical distributed atoms of P-NiMoHZ (Fig. 1e) are consistent with the (110) plane of the β-NiMoO₄ unit cell model. When using these theoretical crystal models to simulate XRD patterns, the obtained peaks match well with the experimental ones (Supplementary Fig. 10), further indicating that the structures are in high conformity with the crystal models. Surprisingly, from the amplified HAADF-STEM image of P-NiMoHZ (Supplementary Fig. 11), in addition to the regular configuration, some darkened sites could be observed, which is due to the loss of the original atoms in those sites. Since the bright dots represent the metal atoms[41], according to the analysis of XPS, the relatively dark site marked by the red arrow might be the substitution of Mo atom by P atom, which results from the replacing of Mo–O tetrahedron by PO₄³⁻ tetrahedron. It could be approved by the intensity profile of HAADF-STEM as well. As shown in Fig. 1f, when P atoms occupy the sites of metal atoms, the intensity undergoes a sudden decrease due to the smaller atomic size of phosphorus compared to nickel and molybdenum. Besides, considering the high conformity between atomic arrangement and theoretical plane model, the gloomy sites marked by the orange arrows between bright dots might be ascribed to the Vₒ (Supplementary Fig. 11). Moreover, transmission electron microscopy (TEM) image and selected area electron diffraction (SAED) image analysis confirm that the crystal structure of P-NiMoHZ is β-NiMoO₄ (Supplementary Fig. 12a). From HAADF-STEM image (Supplementary Fig. 12b) and corresponding elemental mapping (Fig. 1g), P-NiMoHZ is

composed of C, N, Ni, Mo, O, and P elements, a reasonable result based on the structure of P-NiMoHZ. The line scans profile (Supplementary Fig. 12b) exhibits that the signals of C and N derived from the carbonization of HZIF are stronger than the rest four elements at the edge of the nanorod, supporting the core-shell structure of NiMoHZ. Significantly, the same signal positions of P, Ni, Mo, and O suggest that P enter into the lattice of β-NiMoO₄ rather than the outer carbon layer.

**Phase transformation mechanism.** Based on the above analysis, it is known that thermodynamically metastable β-NiMoO₄ could be maintained at ambient temperature with the existence of phosphate groups and oxygen vacancies. To further elucidate the critical roles of PO₄³⁻ and Vₒ in phase transformation process, a series of control groups were prepared and characterized. As shown in Supplementary Fig. 13, the XRD pattern of the C-NiMoHZ (without feeding of phosphate, m = 0 mg) belongs to the standard diffraction of the α-NiMoO₄, while those with different amount of NaH₂PO₂·H₂O (m = 30, 60, 90, 120 mg) and different reaction time (t = 30, 60, 90 min) all correspond well to the diffraction of β-NiMoO₄. Meanwhile, the element atomic percent contents of O, Mo, Ni, P, N of control groups were determined by energy-dispersive X-ray spectroscopy (Supplementary Table 1). From the EPR spectra and phosphorus atomic content comparison (Supplementary Figs. 14, 15), it is clear that β-NiMoO₄ could not be achieved without the existence of

$PO_4^{3-}$ (e.g. C-NiMoHZ with $V_o$), indicating a more significant role of $PO_4^{3-}$ for phase stabilization. Moreover, the more phosphate groups in material could induce more oxygen vacancies. To this end, oxygen vacancies (positive charge) are deemed to balance the charge since $PO_4^{3-}$ is more negative than $MoO_4^{2-}$ when it occupies the place of Mo–O tetrahedrons[13]. Theoretical calculations of the metal vacancy formation energy in β-NiMoO$_4$ (Supplementary Fig. 16) suggest that the Mo vacancy compound model is thermodynamically more stable than Ni vacancy, which is more favorable for the substitution of phosphate. To confirm this hypothesis, three typical samples (CP-NiMoHZ, RP-NiMoHZ, and P-NiMoO$_4$, details see in experiment sections and Supplementary Fig. 2) were designed. The corresponding XRD patterns in Supplementary Fig. 17 demonstrate that α-NiMoO$_4$ could be converted into β phase by further phosphating reaction (C-NiMoHZ to CP-NiMoHZ; NiMoO$_4$ to P-NiMoO$_4$), verifying the indispensable role of phosphate. From EPR result, the existence of $V_o$ in P-NiMoO$_4$ could be proved (Supplementary Fig. 18). On the other hand, the partial transformation from β phase back to α phase (RP-NiMoHZ) after oxidative treatment highlights the subordinate function of $V_o$. Hence, according to all these crystal evolution analysis and structural characterizations, metastable β-NiMoO$_4$ transformation mechanism was proposed. At an appropriate temperature, α-NiMoO$_4$ would automatically transform into β-NiMoO$_4$[29]. Then, $PO_4^{3-}$ enters into the lattice of NiMoO$_4$ and takes the place of M-O tetrahedron in the crystal structure of β-NiMoO$_4$ during phosphating (phosphate substitution, as shown in Fig. 2a), at the same time devoting to the thermodynamic optimization of the formation energy.

**Electronic state analysis of NiMoO$_4$.** As shown in Fig. 2b, after the phosphate treatment, there is a slight shift to higher binding energy of the characteristic Ni $2p_{1/2}$ and $2p_{3/2}$ peaks in P-β-NiMoO$_4$, which imply that the partial charge of Ni atom after phosphate treatment are transferred to neighboring atoms such as P and O atom[5]. And that means there are some optimizing

electronic states originated from Ni atom near the Fermi level in P-β-NiMoO$_4$ compared to the α-NiMoO$_4$ system, which facilitates the charge transfer from activated Ni to surrounding atoms (Fig. 2c). Similar to that of previous work[15,42,43], the generation of active electron states are mainly attributed to the uplifting the states of Ni-$3d$ after the phosphate substitution in β-NiMoO$_4$ system. Thus, the adsorbed protons could easily receive the electrons to produce hydrogen atoms, thus accelerating the whole HER process on P-β-NiMoO$_4$.

**HER performance measurements.** The HER performance of Pt/C, P-NiMoHZ, P-NiMoO$_4$, C-NiMoHZ, NiMoO$_4$ and other as-prepared electrocatalysts were evaluated in 1 M KOH electrolyte via a conventional three-electrode system[44–46]. Polarization curves (Fig. 3a) display that P-NiMoHZ requires a low overpotential of only −23 mV at 10 mA cm$^{-2}$, a value even lower than that of the Pt/C electrode (−38 mV). Furthermore, P-NiMoHZ shows a quite stable catalytic performance at the current density of 10 mA cm$^{-2}$ during 10 h chronopotentiometry measurement (Supplementary Fig. 19a). The overpotential for P-NiMoO$_4$, C-NiMoHZ, and NiMoO$_4$ at 10 mA cm$^{-2}$ was −91 mV, −151 mV, and −206 mV, respectively, which is in line with the theoretical prediction of active electronic states. As expected, pure P-β-NiMoO$_4$ (P-NiMoHZ) possessed the best HER activity, while the activity of rest samples decreased with the rising ratio of α-NiMoO$_4$. The better performance of C-NiMoHZ than NiMoO$_4$ is due to its shell layer derived from the carbonization of the HZIF, which could increase the conductivity of the electrode to some extent[47]. The values of hydrogen evolution rate for these electrocatalysts follow the same trend as well, among which P-NiMoHZ has the highest rate of 16640 mL g$^{-1}$ cm$^{-2}$ h$^{-1}$ (Supplementary Table 2), while the corresponding Faradic efficiency can be as high as 99%. Besides, electrocatalytic performances of other control groups were measured as well. From the results of polar curve (Supplementary Fig. 19b–d), a positive correlation relationship was

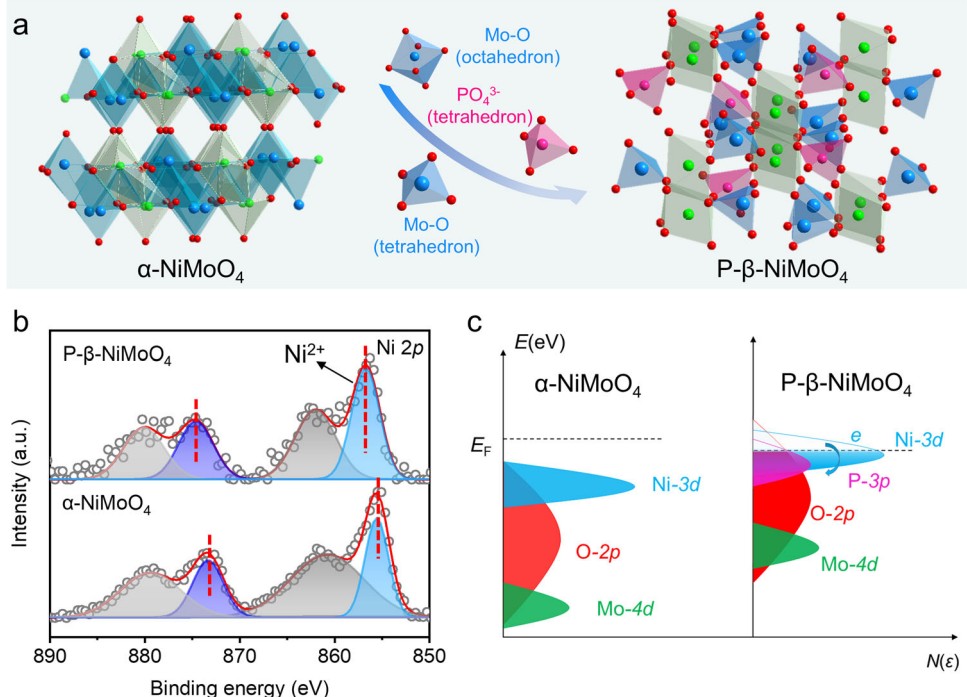

**Fig. 2 Crystal structures and electronic states of α-NiMoO$_4$ and P-β-NiMoO$_4$. a** Schematic of crystal structure evolution during phosphate substitution. **b** XPS spectra of Ni $2p$ in P-β-NiMoO$_4$ and α-NiMoO$_4$. **c** Schematic illustration of the generation of active electric states in different phase NiMoO$_4$.

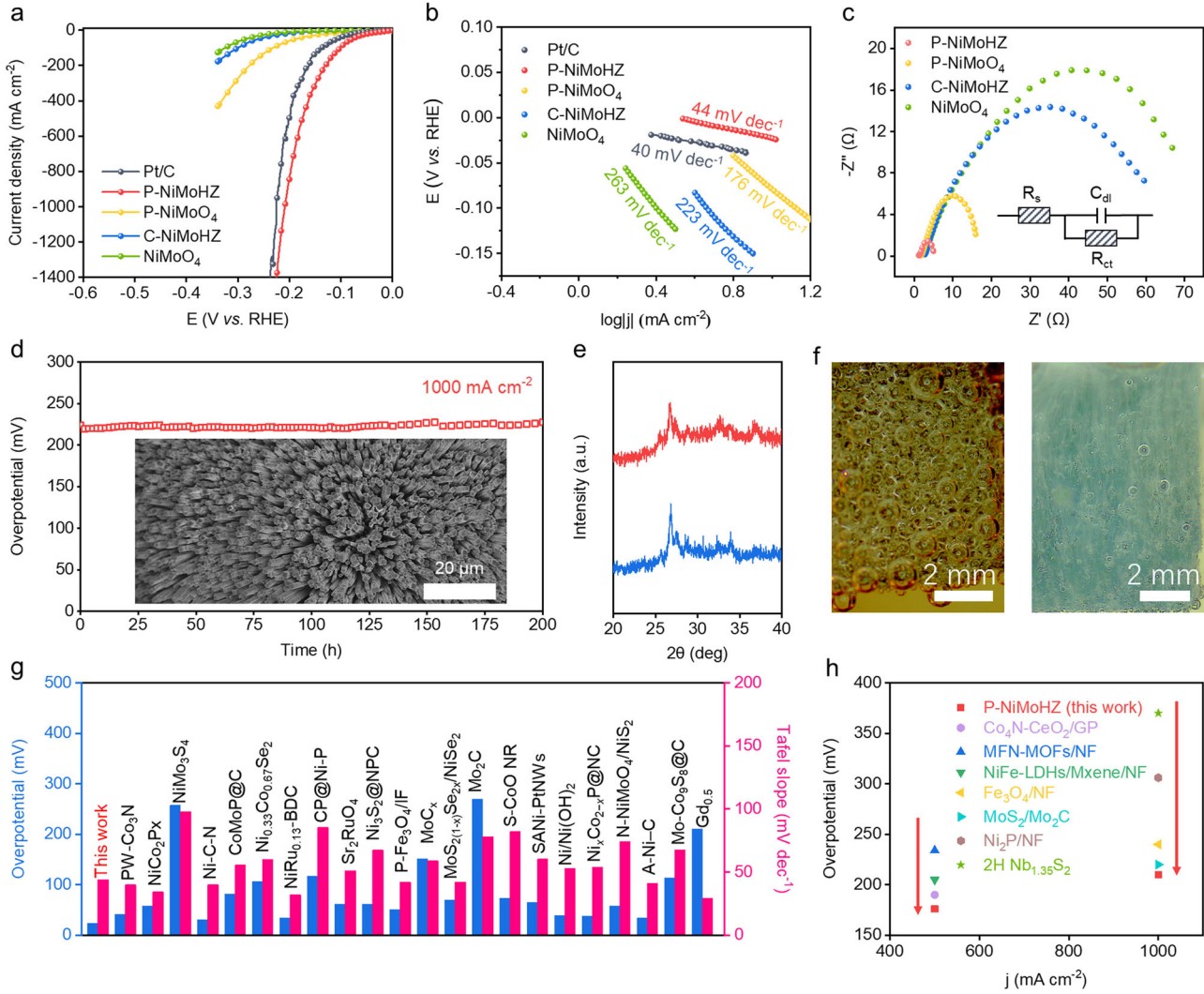

**Fig. 3 Electrocatalytic properties for HER. a** Polarization curves of NiMoO₄, C-NiMoHZ, P-NiMoO₄, P-NiMoHZ, and Pt/C in 1.0 M KOH saturated with $N_2$ gas at a scan rate of 5 mV s⁻¹. **b** The corresponding Tafel plots. **c** Nyquist plots. The inset is the equivalent circuit schematic. **d** Long-term stability tests of P-NiMoHZ at large current density (1000 mA cm⁻²). Inset: the SEM image of P-NiMoHZ after the stability test. **e** XRD patterns of the P-NiMoHZ before and after the stability test. **f** Photographs of the electrode during HER electrocatalysis (left: NiMoO₄; right: P-NiMoHZ). **g, h** The comparison of overpotentials and Tafel slopes of P-NiMoHZ and other reported catalysts at small (**g**) or large current density (**h**).

found between the catalytic activity, the amount of phosphate substitution, and the quantity of $V_o$, however, except in the cases of P-120 mg and P-90 min. It was speculated that excessive phosphate substitution and $V_o$ amount might cause a distortion in the local crystal structure[5], which would reduce the electron transfer efficiency, thus decreasing the catalytic performance. By comparison, the structure and the HER activity of the other two non-metal-doping electrocatalysts (N-NiMoHZ and S-NiMoHZ) were investigated (Supplementary Figs. 20–22).

To further evaluate the HER kinetic mechanism of these electrocatalysts, Tafel slope, a direct indicator of kinetic mechanism, is calculated and plotted in Fig. 3b[48,49]. As expected, the Tafel slope of P-NiMoHZ was only 44 mV dec⁻¹, which was very close to the Pt/C (40 mV dec⁻¹) and far lower than P-NiMoO₄ (176 mV dec⁻¹), C-NiMoHZ (223 mV dec⁻¹), and NiMoO₄ (263 mV dec⁻¹), indicating the Volmer-Heyrovsky mechanism as the HER pathway[50,51]. For P-NiMoHZ and Pt/C, the Heyrovsky step is the rate-determining step, while the Volmer step is rate-determining for the rest electrocatalysts. This result verifies the prediction of active electronic state calculation that P-NiMoHZ can accelerate the rate of hydrogen atom formation

(Volmer step). Besides, the Tafel slopes of other control groups (Supplementary Fig. 23) indicate the same reaction mechanism but with different rate-determining steps. The electrochemical surface area (ECSA) is another important factor for the evaluation of electrocatalysts[40]. Since there is a defined relationship between the electrochemical double-layer capacitance ($C_{dl}$) and ECSA (the higher $C_{dl}$, the larger ECSA), the cyclic voltammetry (CV) measurement was applied to determine the $C_{dl}$ of each electrocatalyst[27]. According to the CV curves (Supplementary Figs. 24–26), the calculated $C_{dl}$ value of P-NiMoHZ was the largest (Supplementary Fig. 27), implying its highest exposure area of the active sites. Furthermore, electrochemical impedance spectroscopy (EIS) was conducted to better probe the charge transfer process towards HER. The charge transfer resistance ($R_{ct}$) of P-NiMoHZ was ~3 Ω versus ~14 Ω for P-NiMoO₄, ~57 Ω for C-NiMoHZ, ~64 Ω for NiMoO₄. EIS values of the other control samples were presented in Supplementary Fig. 28a. Obviously, the smallest $R_{ct}$ of P-NiMoHZ indicates the most favorable kinetic of charge transfer, therefore a faster reaction rate of HER. In addition, the turnover frequency (TOF) values were calculated to evaluate the intrinsic activity of

P-NiMoHZ. Surprisingly, it shows a high TOF value of 0.76 s⁻¹ at the overpotential of 100 mV, which are larger than those of the other reported state-of-the-art HER electrocatalysts in 1 M KOH in Supplementary Table 3, representing its excellent intrinsic activity.

**Activity and stability at large current density**. The stability at large current density is a critical standard to evaluate an electrocatalyst's feasibility in industrial applications[52,53]. Since P-NiMoHZ could reach a very high current density at relatively low overpotential (Fig. 3a), its long-term stability at a large current density of 1000 mA cm⁻² was tested. As shown in Fig. 3d, there was no apparent degradation of overpotential even after the 200 hours' test, which demonstrates the potential as an excellent electrocatalyst for industrial applications. Moreover, compared to the initial polarization curve, the HER activity of P-NiMoHZ exhibited little variation after the stability test (Supplementary Fig. 28b), further indicating its superior stability. Whatever at small or large current densities, P-NiMoHZ displays better HER activity compared to other state-of-the-art electrocatalysts (Fig. 3g, h, and Supplementary Table 4–6).

Basically, the high activity originates from the properties of materials and structures (eg. P-β-NiMoO₄ and fine-nanorod arrays). Accordingly, their evolution is the key to unveil the origin of the high electrocatalytic stability of P-NiMoHZ. Firstly, from the polarization curve of P-NiMoHZ in Fig. 3a, no redox peak is observed. That means P-β-NiMoO₄ is still the active material during the HER electrocatalysis, and no extra reaction is taking place except for HER. The XRD patterns of P-NiMoHZ before and after stability test (Fig. 3e) also confirm the stability of this material. Besides, more attention was paid to the surface states of P-NiMoHZ to ascertain whether there is surface reconstruction in the process of HER electrocatalysis[54,55]. The in-situ/ex-situ Raman spectroscopy and XPS (Supplementary Figs. 29 and 30) demonstrate that the surface structure of the electrocatalyst remains unchanging. In other words, no phase or material comes into being, namely, no obvious surface reconstruction. Furthermore, the electrolyte was collected timely during the long-term stability test to clarify whether there was a dissolution of the active material on the electrode. Through the analysis of the results from the inductively coupled plasma-mass spectrometry (ICP-MS) (Supplementary Fig. 31b), it is clear that no material dissolution took place. SEM images after stability test (inset of Fig. 3d and Supplementary Fig. 32b) show that the nano-morphology (structure) is well preserved, representing that the active site amount is not influenced. On the other hand, the bubble effects and hydrophilicity were further investigated to interpret the high stability. From the contact angle measurements (Supplementary Movie 1, 2 for P-NiMoHZ, Supplementary Fig. 33), in contrast to NiMoO₄, P-NiMoHZ is hydrophilic and much more gas-phobic. Therefore, from macroscopic view, during HER process, the produced H₂ gas bubble will be easily released on the electrode surface, at the same time the surface will quickly get wet by water again. The overall effect is that at the same current density, the size of bubbles on the P-NiMoHZ electrode surface would be much smaller than those on the NiMoO₄ electrode (Fig. 3f), which was confirmed by the bubble size distribution on P-NiMoHZ and NiMoO₄ during HER electrocatalysis (Supplementary Fig. 34). As a result, it will avoid the catalyst shedding issue from drastic large bubble releasing and facilitate the retention of the nanostructure on the electrode surface. Meanwhile, the faster releasing of the bubbles would lead to faster re-exposure of the active sites, which is in favor of electrocatalysis.

**Theoretical simulation**. To understand the primarily rate limiting step of the HER with H₂O reduction, we present an in-depth discussion regarding the relationship between hydrogen adsorption behavior and the water dissociation process, and their corresponding roles in the overall alkaline HER rate. DFT calculations were performed to investigate a cooperative catalytic mechanism of P-β-NiMoO₄ (P-NiMoHZ) system. For HER in alkaline conditions, there are two continuous steps of water dissociation and hydrogen adsorption[56–58]. For the water dissociation process, it may introduce an additional energy barrier and govern the overall reaction rate. The considerably slow rate of the water dissociation in alkaline electrolyte has greatly hindered the overall high purity hydrogen production and reaction kinetic rate. From the kinetic viewpoints, the activation barriers of water dissociation play an important role in accelerating to provide a neutral hydrogen source. As shown in Fig. 4a, b, it is revealed that the active Ni sites of P-β-NiMoO₄ system (Fig. 4a, b) exhibit much higher adsorption energy (-0.599 eV) for water adsorption, compared with α-NiMoO₄ system (−0.424 eV) (Supplementary Fig. 35). In addition, it is found that there is a linear correlation among H₂O adsorption energy, bond length and the amount of charge transfer Δ $e$. The more active the electron transfer, the stronger the H₂O binding energy, and the shorter the α-NiMoO₄/ P-β-NiMoO₄-H₂O bond length (Supplementary Fig. 36). Thus, the higher adsorption free energy of H₂O corresponds the lower activation barriers of water dissociation. In Fig. 4c, the active Ni sites of P-β-NiMoO₄ (b) system exhibit much lower activation barriers (0.569 eV) than that of α-NiMoO₄ (1.457 eV) and Pt catalysis (0.94 eV) (Supplementary Fig. 37) for water dissociation[59,60]. Consequently, the P-β-NiMoO₄ system could accelerate water dissociation to provide neutral hydrogen source with the lower activation barrier for HER with H₂O reduction. From the microscopic view, the apparent alkaline HER activity is governed by two factors: the lower barrier to water dissociation and appropriate (not too strong nor too weak) hydrogen adsorption. In Fig. 4d, the corresponding charge density differences of H₂O adsorbed on Ni site in P-β-NiMoO₄ are also represented, qualitatively reflecting the redistribution of electron states with the largest amount of charge transfer, further optimizing the decomposition of water.

Besides the H₂O dissociation barrier, the hydrogen adsorption free energy $\Delta G_{H*}$ is also an effective descriptor to estimate the HER activity[61]. Herein, different exposed atomic sites of P-β-NiMoO₄ and α-NiMoO₄ were used to calculate $\Delta G_{H*}$ (Fig. 4e, Supplementary Fig. 38). The O atoms (O1 and O2 site) located around P or Ni atom exhibit an increasing hydrogen adsorption energy, with $\Delta G_{H*}(O1) = -0.046$ eV and $\Delta G_{H*}(O2) = -0.416$ eV, respectively. The optimal HER active site is considered as O1 site connecting P atom and Ni atom, which has appropriate hydrogen binding energy (−0.046 eV) for hydrogen desorption. In general, the different catalytic activity is attributed to oxidization degree of different exposed atoms of P-β-NiMoO₄. As shown in Fig. 4f, the Bader charge calculations show about 0.619 $e$ charge transfer from O1 to hydrogen proton. In comparison, the O atoms connected with Mo and Ni atom in α-NiMoO₄ have less amount of charge transfer than that P-β-NiMoO₄. The structure-property relationship among charge transfer, $\Delta G_{H*}$ and bond length also exhibited linear correlation in the various atoms of different phase NiMoO₄ system (Fig. 4f). Moreover, we also studied the projected density of state (PDOS) of various exposed atoms in α-NiMoO₄ and P-β-NiMoO₄ to understand the origin of high activity (Fig. 4g–i). Through comparing Ni-3$d$, Mo-4$d$ and O-2$p$ electron density below Fermi level between α-NiMoO₄ and P-β-NiMoO₄, it is observed that the amounts of active electron states Ni-3$d$ and O-2$p$ orbitals near Fermi level between −1 and 0 eV remarkably

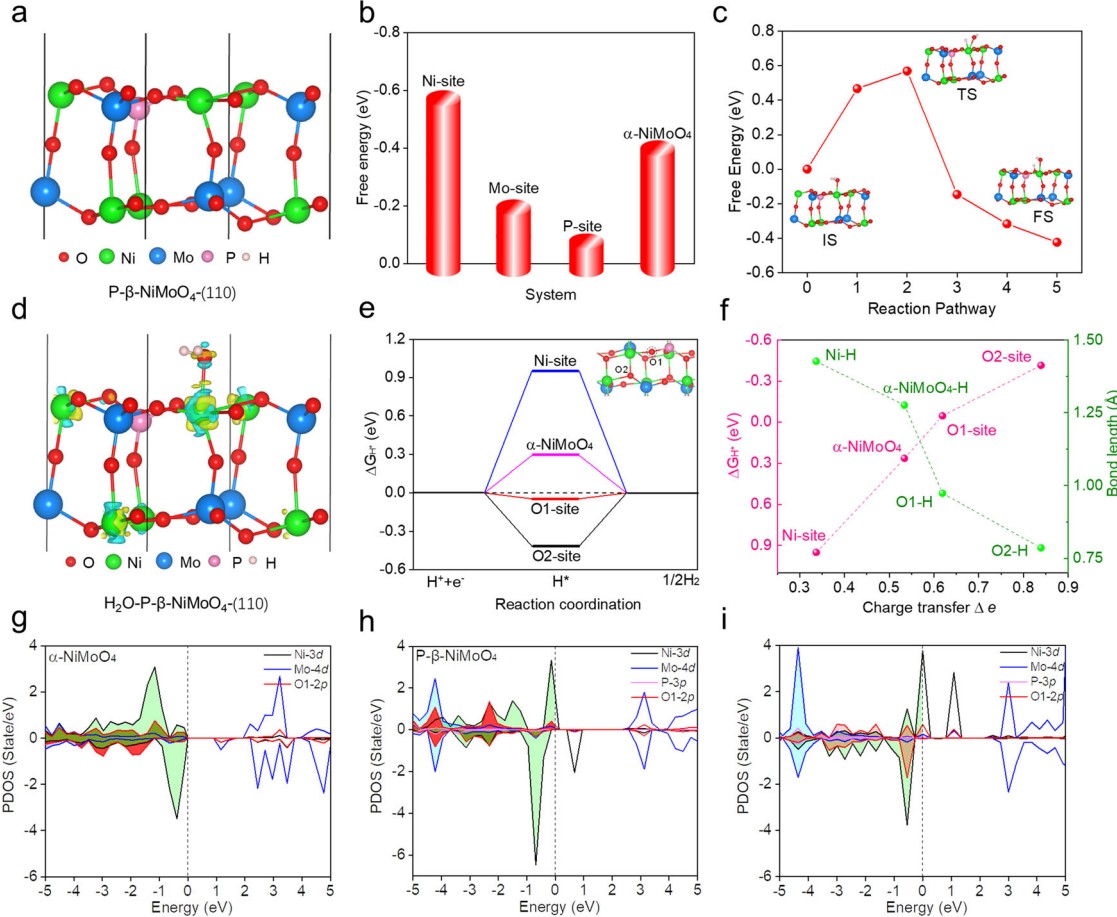

**Fig. 4 Calculated adsorption free energy and electronic structure of different adsorbates between α-NiMoO₄-(110) and P-β-NiMoO₄-(110) surface.**
**a** The optimized structure of P-β-NiMoO₄-(110). Especially, the Mo atom was substituted by the P atom according to the experiment verification. **b** The Gibbs free energy diagram on H₂O adsorbed on different sites in α-NiMoO₄-(110) and P-β-NiMoO₄-(110) surface, respectively. **c** Water dissociation barrier for reaction pathway of P-β-NiMoO₄ system. The insets are the structure of the corresponding IS (initial state), TS (transition state) and FS (final state). **d** The corresponding charge density differences of H₂O adsorbed on Ni sites in P-β-NiMoO₄. The yellow and blue regions indicate the accumulated or dispersed amount of electron states of atoms around the interface, respectively. **e** Hydrogen adsorption free energy (ΔG$_{H*}$) in different exposed atoms in P-β-NiMoO₄-(110). Here, these exposed atoms are considered as the catalytic sites for HER. **f** Linear correlation between ΔG$_{H*}$, P-β-NiMoO₄-H bond length, α-NiMoO₄-H bond length and the amount of charge transfer Δ e of various active sites in different phase of NiMoO₄ surface. **g** Projected density of state (PDOS) of various exposed atoms in α-NiMoO₄-(110). **h**, **i** PDOS before (**h**) and after (**i**) H₂O being adsorbed on the Ni site of P-β-NiMoO₄.

increase corresponding to the intensity of H₂O binding from weak to strong (Fig. 4g–h). In addition, the PDOS before and after H₂O absorbed on P-β-NiMoO₄ system are illustrated the electron states of Ni-3$d$ orbitals near Fermi level between −1 and 0 eV are relatively reduced and partially dragged to the deep energy level (−3 to −2 eV) due to the partially charge transfer and strong hybridization with H₂O-$p$ orbital. Therefore, compared to α-NiMoO₄, the P-β-NiMoO₄ system prefers to regulate the intrinsic charge distribution of exposed atoms, further optimizing the HER performance.

## Discussion

In summary, we developed a facile non-solvent strategy to manipulate the active electronic states of α-NiMoO₄ through phosphate substitution and subsequent phase transformation. Abundant active electronic states in the metastable phase P-β-NiMoO₄ (P-NiMoHZ) were achieved and greatly promote its intrinsic HER activity in alkaline. The phase transformation mechanism unveils that phosphate substitution and oxygen vacancy play a synergistic role in the stabilization of the metastable phase. Furthermore, the significant effects of the

hydrophilic and gas-phobic ability of the electrocatalysts on their stability and kinetics at large current density were highlighted. This study paves an avenue for designing low-cost and highly active non-noble transition metal oxides, especially for HER applications under large current density.

## Methods

**Chemical and materials.** Nickel foam (thickness: 1 mm, porosity: ca. 95%) was bought from Lizhiyuan Ltd. (China). Ni(NO₃)₂·6H₂O, (NH₄)₆Mo₇O₂₄·4H₂O, 2-methylimidazole (2-MIM), NaH₂PO₂·H₂O, C₁₀H₁₆N₂O₈, ethanol, isopropanol (IPA), and 3 M HCl were obtained from Aladdin. Commercial Pt/C (20 wt%) and CH₃C₆H₄SO₃H·H₂O were purchased from Macklin. Nafion solution (~5 wt%) was bought from Alfa-Aesar. All the chemicals were directly used without further purification. Deionized (DI) water (>18.2 MΩ cm) was used to prepare all solutions.

**Synthesis of NiMoO₄·xH₂O and NiMoO₄.** Before being used as the substrate, Ni foam was pretreated by ultrasonication in a 3 M HCl solution for 15 min then followed by washing with DI water and ethanol several times. Afterward, Ni foam was transferred into a 20 mL Teflon-lined autoclave containing 15 mL H₂O, 0.04 M Ni(NO₃)₂, and 0.01 M (NH₄)₆Mo₇O₂₄. The autoclave was sealed and maintained at 160 °C for 6 h. After the reaction, the product was washed with DI water and ethanol several times and dried at 70 °C for 2 h[62]. Dehydrate NiMoO₄ was obtained

by removing the crystal water of $NiMoO_4 \cdot xH_2O$ at 500 °C for 2 h in a tube furnace with Ar atmosphere.

**Synthesis of NiMoHZ.** Sublimation-vapor phase transformation (SVPT) strategy[34] was applied to enable the preparation of core-shell $NiMoO_4$-hybrid zeolitic imidazolate framework (NiMoHZ) on Ni foam. In detail, with 50 mg 2-MIM placed at the bottom of a customized crucible, a piece of $NiMoO_4 \cdot xH_2O$ ($1.0 \times 1.6$ cm$^2$) was put on the hollow quartz tube inside the crucible. Then, the crucible was sealed and maintained in an oven at 180 °C for 2 h (Supplementary Fig. 1).

**Synthesis of P-NiMoHZ.** In a typical process, 90 mg solid $NaH_2PO_2 \cdot H_2O$ was put at the bottom of a ceramic crucible, while NiMoHZ ($1.0 \times 1.6$ cm$^2$) was kept away from the solid powder reactant by vertically putting it on a hollow quartz tube inside the crucible. Afterward, the crucible was put in a tube furnace at 500 °C for 60 min (Ar atmosphere) to get P-NiMoHZ (90 mg, 500 °C, 60 min). During the heating process, $NaH_2PO_2 \cdot H_2O$ would decompose and produce $PH_3$ gas by the following equation:

$$2NaH_2PO_2 \cdot H_2O \rightarrow PH_3 \uparrow + Na_2HPO_4 + 2H_2O \uparrow \quad (1)$$

While keeping the other parameters, the other eight control groups were obtained by changing the reaction temperature (400 °C, 450 °C, and 550 °C), the feed amount of $NaH_2PO_2 \cdot H_2O$ (30 mg, 60 mg, and 120 mg), or the reaction time (30 min and 90 min), and denoted as P-400 °C, P-30 mg, P-30 min, and etc.

**Synthesis of C-NiMoHZ, CP-NiMoHZ, and RP-NiMoHZ.** The synthesis process of C-NiMoHZ was the same as P-NiMoHZ except for no addition of $NaH_2PO_2 \cdot H_2O$. CP-NiMoHZ was fabricated from C-NiMoHZ by further phosphating at 500 °C for 60 min. P-NiMoHZ was retreated in the tube furnace (500 °C, 60 min, air atmosphere) and then cooled to room temperature to get RP-NiMoHZ.

**Synthesis of P-NiMoO$_4$.** The sample was obtained directly from the $NiMoO_4 \cdot xH_2O$ precursor by phosphating (keeping the reaction conditions the same as those of P-NiMoHZ).

**Synthesis of S-NiMoHZ and N-NiMoHZ.** The synthesis process was the same as P-NiMoHZ except for using $CH_3C_6H_4SO_3H \cdot H_2O$ and $C_{10}H_{16}N_2O_8$ as the sulfur source and nitrogen source, respectively[63].

**Synthesis of Pt/C catalyst.** A total of 1 mg Pt/C was added to the mixture of 1 mL IPA and 10 μL Nafion solution (5%). Then, the mixture was sonicated for 12 h to obtain the catalyst ink. Finally, 5 μL catalyst ink was injected on a polished glassy carbon electrode and dried at 80 °C[31,48].

**Material characterizations.** The nano morphology and structure were observed with field emission scanning electron microscopy (FESEM, Tescan MAIA3 XMH), transmission electron microscopy (TEM, JEM-2100, and FEI-Titan Analytical 80-300ST). Energy-dispersive X-ray spectroscopy (EDS) elemental mapping, selected area electron diffraction (SAED), and high-angle annular dark-field scanning transmission electron microscopy (HAADF-STEM) images were also obtained by FEI-Titan Analytical 80-300ST. X-ray diffraction (XRD) patterns were recorded by Bruker D8 Advance X-ray diffractometer (Cu Kα radiation). Electron paramagnetic resonance spectroscopy (EPR) was carried out on Bruker A300 spectrometer. X-ray photoelectron spectroscopy was conducted on PerkinElmer PHI 5000 C. In situ Raman spectroscopy was conducted on the Renishaw inVia Raman Microscope under an excitation of 532 nm laser at a series of controlled potentials by an electrochemical workstation. Ex-situ Raman spectroscopy was conducted on a Dilor LabRAM-1B multichannel confocal micro spectrometer with 514 nm laser excitation. Inductively coupled plasma-mass spectrometry (ICP-MS) was carried out on Agilent 720 instrument. Contact angle was tested by Kruss DY-100 goniometer.

**Electrochemical measurements.** Unless specified, all electrochemical measurements were conducted on the CHI 760E electrochemical workstation (Shanghai Chenhua Instrument Co., Ltd), using a three-electrode system in 1 M KOH electrolyte at room temperature. The as-prepared catalysts were directly applied as the working electrode, while saturated calomel electrode (SCE) and polished graphite rod were used as the reference electrode and the counter electrode, respectively. The working electrode area immersed in the electrolyte was $0.5 \times 0.5$ cm$^2$. Linear sweep voltammetry (LSV) was operated with a scanning rate of 5 mV s$^{-1}$. Polarization curves were corrected with iR compensation (compensation level 85%). Cyclic voltammetry (CV) was conducted at the potential window of −0.60 to −0.70 V (vs. SCE) with a series of scan rates (10, 20, 30, 40, and 50 mV s$^{-1}$). The stability tests were conducted by chronopotentiometry at the current density of 10 mA cm$^{-2}$ for 10 h and 1000 mA cm$^{-2}$ for 200 h, respectively. Electrochemical impedance spectroscopy (EIS) was performed on the Metrohm Autolab PGSTAT302N from 10000 to 0.1 Hz at −0.1 V vs. RHE. The bubble size distributions were observed under chronopotentiometry mode at the current density

of 100 mA cm$^{-2}$. Regarding the 1 M KOH as the electrolyte and SCE as the reference electrode, all potentials were calibrated to the reversible hydrogen electrode (RHE) by the following formula unless specifically noted:

$$E(RHE) = E(SCE) + 0.059 \times pH + 0.2438 \quad (2)$$

**Hydrogen evolution rate and Faradic efficiency test.** To obtain the hydrogen evolution rate and Faradic efficiency (FE)[50], chronopotentiometry was conducted at the current density of 200 mA cm$^{-2}$, while gas chromatograph (Agilent 7890B) was used for $H_2$ gas detection[64]. FE was calculated with the following equation:

$$FE\% = \frac{Q_{hydrogen}}{Q_{total}} \times 100\% = \frac{n \times Z \times F}{it} \times 100\% \quad (3)$$

$Q_{hydrogen}$ is the electric quantity used for hydrogen production. $Q_{total}$ represents the total electric quantity applied. n is the amount of substance in produced $H_2$, while Z is the specific number of electrons required to produce a hydrogen molecule. F is the Faradic constant. i is the current applied and t is the time for applying current.

The hydrogen evolution rate was calculated according to the following formula:

$$\text{hydrogen evolution rate} = \frac{V_{hydrogen}}{m_{cat} \times A_{electrode} \times t} \quad (4)$$

$V_{hydrogen}$ is the volume of produced hydrogen at specific time, while $m_{cat}$ and $A_{electrode}$ are the mass of electrocatalyst on the electrode and the area of the electrode, respectively. t represents the time for gas collection or applying current.

**Computational details.** The spin-polarized density functional theory (DFT) calculations were performed using the Vienna Ab initio Simulation Package (VASP), with the generalized gradient approximation of Perdew–Burke–Ernzerhof to describe electron exchange and correlation[65]. The plane-wave basis cut off was 450 eV. The projector-augmented plane wave (PAW) was used to describe the electron-ion interactions[66]. A set of ($3 \times 3 \times 1$) k-points were carried out for geometric optimization, and the convergence threshold was set as $10^{-4}$ eV in energy and 0.05 eV/Å in force, respectively. The models of α-$NiMoO_4$-(110) and P-β-$NiMoO_4$-(110) were first chosen for simulating the HER performance of different phase of $NiMoO_4$. For the systems, the free energy of the adsorbed state is calculated as:

$$\Delta G_{H*} = \Delta E_{H*} + \Delta E_{ZPE} - T\Delta S \quad (5)$$

Where $\Delta E_{H*}$ is the hydrogen chemisorption energy, and $\Delta E_{ZPE}$ is the difference corresponding to the zero-point energy between the adsorbed state and the gas phase. As the vibration entropy of H* in the adsorbed state is small, the entropy of adsorption of 1/2 $H_2$ is $\Delta S_H \approx -1/2 S_{H_2}^0$, where $S_{H_2}^0$ is the entropy of $H_2$ in the gas phase at the standard conditions.

## Data availability
All data needed to evaluate the conclusions in the paper are presented in the paper and/or the Supplementary Materials. Additional data related to this paper may be requested from the authors. Source data are provided with this paper.

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

## Acknowledgements
This work is financially supported by the National Natural Science Foundation of China NSFC (22133005, 21973107, 51972064), the Program of Shanghai Academic Research Leader (20XD1404100), and the Shanghai Natural Science Foundation of China (21ZR1472900).

## Author contributions
J.C., J.L., M.Y. and J.S. conceived the research. Z.W. and J.C. carried out the synthesis and performed materials characterization and electrochemical measurements. E.S. and J.L. proposed the active electronic states and carried out the theoretical calculations. N.W. conducted the HAADF-STEM measurement. W.Y. and C.W. assisted in the synthesis and characterization of materials. J.C., Z.W., E.S., J.D., Z.X. and P.A. wrote the manuscript. All authors discussed the results and commented on the manuscript.

## Competing interests
The authors declare no competing interests.
