## [Peer Review File · Nature Communications]

Title: Manipulation on Active Electronic States of Metastable Phase β -NiMoO₄ for Large Current Density Hydrogen EvolutionREVIEWER COMMENTS

Reviewer #1 (Remarks to the Author):

This manuscript reports the phase transformation of stable octahedral α -NiMoO₄ to metastable tetrahedral β -NiMoO₄. Diverse activating strategies such as by doping, carbonization are schemed to realize a proficient HER catalyst with ultralow overpotential and high stability. Remarkable results are demonstrated by the phosphorous doped β -NiMoO₄ catalyst. The HER catalyst exhibited impressive results with an overpotential value -23 mV and excellent stability over 200 h at a quite large current density. Further, exclusive experimental study supported by DFT calculations explored the oxygen vacancy-enriched system responsible for superior HER activity. Therefore, I recommend this manuscript to be published in Nature communications after the minor revision.

Detailed comments:

1. Authors should explain the colour codes (light green, yellow and light red) in the schematic illustration in Figure 2.
2. Author should explain the origin and composition of thread like morphology present along with the regular morphology in supplementary Figure 3.
3. Crystallographic diffraction planes should be incorporated in Figure 2 (b).
4. Author should mention the scale of the HR-TEM images in supplementary figure S13 (a).
5. Authors should specify the element percent contents included in supplementary table 1 refer to atomic/weight percentage.
6. Electrochemical impedance spectroscopy (EIS) was recorded at a potential of -0.65 V. Do authors have any justification for selecting the particular potential value?
7. More detailed elaboration on HER kinetic mechanisms of the electrocatalysts based on nature of the Tafel slope should be included.
8. Comprehensive discussions should be proposed regarding the role of oxygen vacancies on HER activity.
9. Determining the electrochemical active surface area (ECSA) from the CV curves ranging a potential window of -0.7 to -0.6 V, particular potential point at which the anodic and cathodic current densities chosen to calculate specific capacitances should be mentioned clearly (Supplementary figure 21).
10. Authors should include a comparison table with the hydrogen evolution rates ($\text{mL g}^{-1} \text{cm}^{-2} \text{h}^{-1}$ or $\text{mmol g}^{-1} \text{cm}^{-2} \text{h}^{-1}$) of the different as-prepared samples.

Reviewer #2 (Remarks to the Author):

Wang et al. describes fabrication of P-doped core-shell NiMoO₄-hybrid zeolitic imidazolate framework for HER applications under large current. The authors need to address following points:

1. Why doping of P shows the beta NiMoO₄ phase?
2. Can you compare other non-metal doping such as nitrogen or others for HER application?
3. Generally, MOF (zeolitic imidazolate framework) suffers from low stability. However, P-doped core-

shell NiMoO₄-hybrid zeolitic imidazolate framework (NiMoHZ) nanorod shows good stability. What are the reasons behind it?

Reviewer #3 (Remarks to the Author):

This study reports highly organized, high surface area electrode based on Ni-Mo for hydrogen evolution reaction. Uniqueness of the study is to synthesize Ni-Mo-O species via sublimation-vapor phase transformation (SVPT) strategy that the authors develop. This leads to the regular rod-shaped arrays on the Ni foam substrate. The obtained electrode shows high current with low overpotential with good stability for HER, even at high current density region. The combination of Ni-Mo has already been applied for long time in alkaline electrolyzer, and there is boosting fashion of reporting this material in recent years. Such recent activities include *Angew. Chem. Int. Ed.* 2021, 60, 7051, *Angew. Chem. Int. Ed.* 2021, 60, 5771. Basically, what is missing in this article is correct description of how the experiments were conducted, and how the calculations were introduced to identify which properties of the materials. To my view, the paper is poorly written, and it is not ready for publication. NiMo oxide should be (at least partly) reduced upon HER condition. Scientific content, if any, should be well improved with recent understanding of this particular material.

- First, general scientific narrative should be improved, which reflects the quality of the science of the authors. For example, Abstract starts with "Transition Metal oxides (TMOs) are abundant in nature", where noble metal oxides including Pt and Ir oxides are also TMOs.
- The paper is written in an unintellectual order of messages. Results start with density of state calculation, before even talking about how "P-doping" is achieved and where the location of P species is located. How can the readers be convinced with the calculation? Even if it is accepted, what is the evidence that PO₄ substitution makes metallic character (half filled) of Ni? What is the size of the cell to obtain this?
- For this Figure 1, after reading carefully the latter part of study, P seems to be present as phosphate, which is reasonable, but what was the evidence to substitute MoO₄ site? Just having the same structure is not sufficient.
- Also I recommend the appropriate use of P doping and phosphate substitution. P doping is typically used for semiconductor (such as Si). This is P substitution as Fig. 1 caption says, but more specifically phosphate substitution.
- Related to above phosphate substitution, it is unclear how the experiment is conducted. Solid hypophosphite is mixed in crucible? How reproducible is this experiment?
- Recent understanding of Ni-Mo species for HER is that the electrode is "alive" upon the potential sweeping: the dissolution and reorganization of Mo species on the surface happens. While NiMo alloy is commonly being accepted NOT to be the active site, but more and more papers discuss the crucial role of Mo oxides on the surface as the true active redox species. See, e.g., *ACS Catal.* 2020, 10, 12858.
- Current density of > 1000 mA cm⁻² is reported. This type of measurement requires special care of cell

design as the solution resistance and counter electrode should be adequately optimized. The description of experiment is not sufficient. What is the size of working electrode? At counter electrode, what is the size and how to catch up the 1A with carbon electrode (self oxidation)? It is also questionable how iR correction is achieved if any. If iR correction is not achieved, obtained result is highly skeptical as it should reflect the iR loss in the measured potential. If iR correction is done, slight difference in this value would change the description of the performance. There should be bubble problem of the H₂ product. What is the power source used to achieve this high current range?

- For capacitance correlation to calculate TOF, it is true that many papers discuss this, but my personal view is not essential. Highly porous, high surface area material, ECSA would underestimate the active surface. This material shows high current density per geometric area, so that the

- Towards the latter part, the discussion starts to talk about H₂O adsorption etc. But it is commonly considered that the alkaline HER is pinned by the rate controlling O-H bond dissociation of H₂O.

Although the thermodynamic calculation discussed in this study is commonly done, it is irrelevant for alkaline HER. For example, H₂O adsorption is not the rate determining step at all.

Overall, there are a lot of papers investigating Ni-Mo species for HER. This study provides (when provided in more details) a protocol how to increase the surface area, enhanced even more with defective structure introduced by hypophosphite species. Performance is as good as other recent similar papers, and worth being reported, but my recommendation is to publish in electrochemistry specific journals because this study does not provide (or even mislead) new scientific content of the active site.

Point-by-point response letter

Reviewers' comments:

Reviewer #1 (Remarks to the Author):

This manuscript reports the phase transformation of stable octahedral α -NiMoO₄ to metastable tetrahedral β -NiMoO₄. Diverse activating strategies such as by doping, carbonization, are schemed to realize a proficient HER catalyst with ultralow overpotential and high stability. Remarkable results are demonstrated by the phosphorous doped β -NiMoO₄ catalyst. The HER catalyst exhibited impressive results with an overpotential value -23 mV and excellent stability over 200 h at a quite large current density. Further, exclusive experimental study supported by DFT calculations explored the oxygen vacancy-enriched system responsible for superior HER activity. Therefore, I recommend this manuscript to be published in Nature communications after the minor revision.

Response: We sincerely thank the reviewer's positive comments. The proposed suggestions are valuable and helpful for improving our paper.

We have carefully revised the manuscript (changes were marked by **highlighted**) and replied to the comments point-by-point shown below.

Detailed comments:

1. Authors should explain the colour codes (light green, yellow and light red) in the schematic illustration in Figure 2.

Response: Thanks for the reviewer's advice. Relevant annotation has been added in the caption of **Figure 1** (Figure 2 in our previous manuscript).

Fig. 1. Design and structure characterization of α -NiMoO₄ and β -NiMoO₄. Schematic illustration of the preparation process for P-NiMoHZ. The light green, light red, and yellow balls represent Ni²⁺ cations, MoO₄²⁻ anions, and NiMoO₄ crystals, respectively. The ball-and-stick models around nanorod arrays are vaporized 2-MIM (middle) and PH₃ (right) molecules.

2-MIM: 2-methylimidazole; PH_3 : phosphine; NiMoHZ: NiMoO_4 -hybrid zeolitic imidazolate framework; P-NiMoHZ: phosphated NiMoHZ.

2. Author should explain the origin and composition of thread like morphology present along with the regular morphology in Supplementary Figure 3.

Response: Thanks for the reviewer's suggestion.

(1) According to our previous work, it was found that during the sublimation-vapor phase transformation (SVPT) process, the morphology evolution undergoes three stages (surface reaction, seed formation, and crystal growth) [*Adv. Funct. Mater.* **29**, 1903875 (2019)]. As shown in Figure R1, at the third stage, the metal-organic framework (MOF) seeds, products of the reaction between oxide and vaporized ligand, will further grow and become bigger crystals.

Figure R1. Schematic illustration of the growth process of MOF crystals by SVPT

In this work, we also applied this strategy to prepare oxide-MOF intermediate (NiMoHZ) by using $\text{NiMoO}_4 \cdot \text{H}_2\text{O}$ and 2-methylimidazole (2-MIM) ligand. Additional explanations are added in the revised SI as follows:

Supplementary Figure 3. SEM images of (a) precursor $\text{NiMoO}_4 \cdot x\text{H}_2\text{O}$ and transformed NiMoHZ with different reaction time: (b) 1 h, (c) 2 h, (d) 3 h.

SI Page 7: “Basically, after 2 h reaction, thin MOF coating forms on the surface of nanorod NiMoO_4 (Supplementary Fig.3c). However, when prolonging the reaction time to 3 h, the coating further grows into thread-like MOF crystals (Supplementary Fig.3d), which is consistent with the previously reported work. [*Adv. Funct. Mater.* **29**, 1903875 (2019)] In order to keep the fine nanorod structure, the 2 h sample was, therefore, chosen as intermediate precursor for the following NiMoO_4 crystal phase conversion and named NiMoHZ.”

(2) The composition of these samples was determined by XRD. Additional explanations are added in the revised SI as follows:

SI Page 8: “According to the results of SEM and XRD in Supplementary Fig. 3 and 4, the structure of NiMoHZ was determined as hydrated NiMoO_4 (inner nanorod) and hybrid zeolitic imidazolate framework (HZIF, outer thin coating) composite.”

Supplementary Figure 4. XRD patterns of $\text{NiMoO}_4 \cdot x\text{H}_2\text{O}$, NiMoHZ, NiMoHZ powder, and 2-MIM powder. The peak in green marked area belongs to $\text{HZIF}(\text{Ni}_4(\text{im})_6\text{MoO}_4)$.

3. Crystallographic diffraction planes should be incorporated in Figure 2 (b).

Response: We appreciate the reviewer’s suggestion. Four main crystallographic diffraction planes of P-NiMoHZ and five main planes of NiMoO_4 have been incorporated and annotated in Figure 1 (b) (Figure 2 (b) in our previous manuscript) as follows:

Fig. 1. Design and structure characterization of α -NiMoO₄ and β -NiMoO₄. b XRD

patterns of P-NiMoHZ and NiMoO₄.

NF: nickel foam; NiMoHZ: NiMoO₄-hybrid zeolitic imidazolate framework; P-NiMoHZ: phosphated NiMoHZ.

4. Author should mention the scale of the HR-TEM images in Supplementary Figure S13 (a).

Response: Thanks for the reviewer's careful checking. We have added the scale bars in Supplementary Fig. 12a (Supplementary Fig. 13a in our previous manuscript) as follows:

Supplementary Figure 12. (a) TEM, HR-TEM and SAED image of NiMoHZ; (b) Cross-sectional elemental distributions by line scans along the white line.

5. Authors should specify the element percent contents included in supplementary table 1 refer to atomic/weight percentage.

Response: Thanks for the reviewer’s suggestion. We apologize that in our previous manuscript the related description was not explicit. We have revised the caption of Supplementary Table 1 and the description in the manuscript as follows:

Manuscript page 9: “Meanwhile, the element **atomic percent** contents of O, Mo, Ni, P, N of control samples were determined by EDX (Supplemental Table 1).”

SI page 39:

Supplementary Table 1. Elemental **atomic** percent contents of all control groups.

Controlled groups	O (%)	Mo (%)	Ni (%)	P (%)	N (%)	Sum
P-30 mg	33.04	32.04	31.29	2.64	1.00	100
P-60 mg	46.17	24.97	23.89	3.03	1.94	100
P-NiMoHZ	61.79	11.60	13.94	10.73	1.95	100
P-120 mg	49.63	16.14	13.62	16.77	3.84	100
P-30 min	51.84	20.91	18.26	5.03	3.96	100
P-90 min	43.54	14.76	24.79	14.11	2.81	100
P-400 °C	62.06	16.92	18.17	2.8	0.06	100
P-450 °C	58.08	16.81	16.26	7.66	1.19	100

6. Electrochemical impedance spectroscopy (EIS) was recorded at a potential of -0.65 V. Do authors have any justification for selecting the particular potential value?

Response: Thanks for the reviewer’s comment.

(1) We are sorry that our clerical error (-0.65 V) in the original text misled the reviewer. Actually, we set the voltage of -0.1 V vs. reversible hydrogen electrode (RHE) to test the EIS. The selection of the overpotential of 0.1 V is based on the valid working voltage for hydrogen evolution reaction (HER) since at this voltage HER and intense charge transfer take place on the electrode. [*Nat. Commun.* **11**, 2940 (2020)]. The voltage of -0.65 V vs. saturated calomel electrode (SCE) is for the calculation of electrochemical double-layer capacitance (C_{dl}) since we chose the range of -0.6 to -0.7 V vs. SCE to record the cyclic voltammetry (CV) curves (more description about CV and C_{dl} can also be found in the response of **Question 9**).

(2) We have revised the electrochemical measurements section as well as the corresponding description in the highlighted manuscript as follows:

Electrochemical measurements in manuscript page 25: “Electrochemical impedance spectroscopy (EIS) was performed on the Metrohm Autolab PGSTAT302N from 10000 to 0.1 Hz at -0.1 V vs. RHE.”

7. More detailed elaboration on HER kinetic mechanisms of the electrocatalysts based on nature of the Tafel slope should be included.

Response: Thanks for the reviewer’s constructive suggestion.

(1) In principle, the mechanism of the HER in alkaline media (as we chose 1 M KOH as the electrolyte) involves three elementary steps: the Volmer step—water dissociation and formation of a reactive intermediate H_{ad} ($2H_2O + M + 2e^- \rightarrow 2M-H_{ad} + 2OH^-$)—followed by either the Heyrovsky step ($H_2O + H_{ad}-M + e^- \rightarrow M + H_2 + OH^-$) or the Tafel recombination step ($2M-H_{ad} \rightarrow 2M + H_2$). [*Science* **334**, 1256-1260 (2011)] The HER pathway can follow Volmer-Heyrovsky step or Volmer-Tafel step as shown above. As an indicator of HER kinetic mechanisms, Tafel slope can be utilized to distinguish the rate-determining step by comparing it to the theoretical values of three elementary steps. [*Nat. Commun.* **12**, 3021 (2021); *Chem. Soc. Rev.* **45**, 1529–1541 (2016)]

(2) We have carefully revised the manuscript and made a more detailed discussion by referring to authoritative works as follows:

Manuscript page 13-14: “To further evaluate the HER kinetic mechanism of these electrocatalysts, Tafel slope, a direct indicator of kinetic mechanisms, was calculated and plotted in Fig. 3b. As expected, the Tafel slope of P-NiMoHZ was only 44 mV dec⁻¹, which was very close to the Pt/C (40 mV dec⁻¹) and far lower than P-NiMoO₄ (176 mV dec⁻¹), C-NiMoHZ (223 mV dec⁻¹), and NiMoO₄ (263 mV dec⁻¹), indicating the Volmer-Heyrovsky mechanism as the HER pathway. [*Nat. Commun.* 2020, 11, 1029; *Nat. Commun.* 2020, 11, 2940] For P-NiMoHZ and Pt/C, the Heyrovsky step is the rate-determining step, while the Volmer step is rate-determining one for the rest of electrocatalysts. This result verifies the prediction of active electronic state calculation that P-NiMoHZ can accelerate the rate of hydrogen atom formation (Volmer step). Besides, the Tafel slopes of the other control groups (Supplementary Fig. 23) indicate the same reaction mechanism but with different rate-determining steps.”

8. Comprehensive discussions should be proposed regarding the role of oxygen vacancies on HER activity.

Response: Thanks for reviewer’s valuable suggestion.

(1) In general, it is accepted that non-metal transition metal oxides (TMOs) are catalytically inert for HER due to their inappropriate hydrogen adsorption strength and poor conductivity. [*Nat. Commun.* **5**, 4695 (2014); *Chem. Soc. Rev.* **49**, 9154–9196 (2020)] However, oxygen vacancy (V_o) was found to be efficient in the modulation of the material electronic structure and reduction of the protons or H_2O adsorption energy barrier on adjacent atoms instead of serving itself as the active site. [*Adv. Mater.* **31**, 1807771 (2019); *Nano Lett.* **17**, 7968–7973 (2017)]

In this work, V_o (positive charge) balances the charge since PO_4^{3-} is more negative than MoO_4^{2-} when it occupies the place of Mo-O tetrahedrons. As shown in Supplementary Figure 15, the more content of PO_4^{3-} , the more amount of V_o .

Supplementary Figure 15. (a) EPR spectra of P-NiMoHZ control groups with different amount of $NaH_2PO_2 \cdot H_2O$ (left) and different reaction time (right). (b) Phosphorus atomic percent of these control groups. EPR spectra show a positive correlation relationship between the V_o and the P atomic content in control groups.

(2) The relation between oxygen vacancy and HER activity was added in the revised manuscript and SI below the Supplementary Fig. 19 as follows:

SI page 23: “P atomic percent and V_o amount order: P-30 mg < P-60 mg < P-90 mg < P-120 mg; P-30 min < P-60 min < P-90 min. (From the results in Supplementary Fig. 15) HER activity order: P-30 mg < P-60 mg < P-120 mg < P-90 mg; P-30 min < P-90 min < P-60 min.

Based on the theoretical calculation results (Fig. 2c and Fig. 4), the active sites are the Ni atoms for H_2O dissociation and O1 sites connected with P and Ni for hydrogen desorption. Similar to previously reported works [*Angew. Chem. Int. Ed.* **60**, 14117–14123 (2021); *Adv. Mater.* **31**, 1807771 (2019); *Nano Lett.* **17**, 7968–7973 (2017)], V_o

is considered to adjust the electronic structure of β -NiMoO₄ and increase its conductivity. Therefore, when increasing the phosphating degree, the V_o amount would simultaneously increase, thus both contributing to the great enhancement of the active site number and conductivity, while the crystal structure remains unchanged. As a result, the HER performance would accordingly be improved.”

Supplementary Figure 19. (a) Chronopotentiometry curve of P-NiMoHZ for 10 h at 10 mA·cm⁻². HER polarization curves of P-NiMoHZ control groups with (b) different temperature, (c) different amount of NaH₂PO₂·H₂O, and (d) different reaction time. The curves of 500 °C, 90 mg, and 60 min were the same, which all belong to the optimal P-NiMoHZ sample.

(3) A positive correlation relationship was found between the catalytic activity, the amount of phosphate substitution, and the quantity of V_o , however, except in the cases of P-120 mg and P-90 min. It was speculated that excessive phosphate substitution and V_o amount might cause a distortion in the local crystal structure, which would reduce the electron transfer efficiency, thus decreasing the catalytic performance. These reverse changes were approved by the C_{dl} and EIS results as well. Further analysis was added in the SI below the Supplementary Fig. 28.

SI page 32: “ C_{dl} values (Supplementary Fig. 27b, c) and EIS results (Supplementary Fig. 28a) of P-NiMoHZ control groups approve the relationship among active site

number, conductivity, phosphating degree, and V_o amount until they reached the best HER performance. Then P-120 mg and P-90 min experienced a reverse change due to the excessive phosphate substitution and following local crystal distortion.”

Supplementary Figure 27. C_{dl} values of P-NiMoHZ control groups with different (b) reaction time, (c) amount of $\text{NaH}_2\text{PO}_2 \cdot \text{H}_2\text{O}$.

Supplementary Figure 28. (a) Nyquist plot of electrocatalysts P-NiMoHZ and control groups.

9. Determining the electrochemical active surface area (ECSA) from the CV curves ranging a potential window of -0.7 to -0.6 V, particular potential point at which the anodic and cathodic current densities chosen to calculate specific capacitances should be mentioned clearly (Supplementary Figure 21).

Response: Thank the reviewer’s suggestion. The relevant description has been added below the Supplementary Fig. 27 (Supplementary Fig. 21 in our previous manuscript) as follows:

SI page 31: “Since the potential window of -0.6 to -0.7 V vs. SCE (0.3594 to 0.4594 V vs. RHE, a range with no polarization) was selected to record CV curves, the middle point of 0.4094 V vs. RHE (-0.65 V vs. SCE) was chosen to calculate the difference between the anodic and cathodic current densities at different scan rate. [Nat. Commun. **11**, 2940 (2020); Angew. Chem. Int. Ed. **58**, 12252–12257 (2019)]”

Revision in the electrochemical measurements section as follows:

Manuscript page 25: “Cyclic voltammetry (CV) was conducted at the potential window of -0.60 to -0.70 V (vs. SCE) with a series of scan rates (10, 20, 30, 40, and 50 mV·s⁻¹).”

10. Authors should include a comparison table with the hydrogen evolution rates (mL g⁻¹ cm⁻² h⁻¹ or mmol g⁻¹ cm⁻² h⁻¹) of the different as-prepared samples.

Response: Thanks for the reviewer’s constructive suggestion.

(1) We have additionally tested the hydrogen evolution rates (mL g⁻¹ cm⁻² h⁻¹) of the four main samples (P-NiMoHZ, P-NiMoO₄, C-NiMoHZ, NiMoO₄) in Figure 3a and tabulated the data as Supplementary Table 2.

SI page 40:

Supplementary Table 2. Hydrogen evolution rate of as-prepared electrocatalysts.

Samples	H ₂ (mL h ⁻¹)	Mass loading (g cm ⁻²)	H ₂ (mL g ⁻¹ cm ⁻² h ⁻¹)
P-NiMoHZ	20.80	0.020	16640
P-NiMoO ₄	13.20	0.022	9600
C-NiMoHZ	10.40	0.020	8320
NiMoO ₄	8.40	0.019	7074

The calculated Faradic efficiency (EF) is as high as 99%.”

Detailed experiment description added in Methods section as follows:

Manuscript page 25: “**Hydrogen evolution rate and Faradic efficiency test.** To obtain the hydrogen evolution rate and Faradic efficiency (FE), chronopotentiometry was conducted at the current density of 200 mA cm⁻², while gas chromatograph (Agilent 7890B) was used for H₂ gas detection. FE was calculated with the following equation:

$$\text{FE \%} = \frac{Q_{\text{hydrogen}}}{Q_{\text{total}}} \times 100\% = \frac{n \times Z \times F}{it} \times 100\%$$

Q_{hydrogen} is the electric quantity used for hydrogen production. Q_{total} represents the total electric quantity applied. n is the amount of substance in produced H_2 , while Z is the specific number of electrons required to produce a hydrogen molecule. F is the Faradic constant. i is the current applied and t is the time for applying current.

The hydrogen evolution rate was calculated according to the following formula:

$$\text{hydrogen evolution rate} = \frac{V_{\text{hydrogen}}}{m_{\text{cat}} \times A_{\text{electrode}} \times t}$$

V_{hydrogen} is the volume of produced hydrogen at specific time, while m_{cat} and $A_{\text{electrode}}$ are the mass of electrocatalyst on the electrode and the area of the electrode, respectively. t represents the time for gas collection or applying current.”

(2) Discussion added in the manuscript as follows:

Manuscript page 13: “The values of hydrogen evolution rate for these electrocatalysts follow the same trend as well, among which P-NiMoHZ has the highest rate of $16640 \text{ mL g}^{-1} \text{ cm}^{-2} \text{ h}^{-1}$ (Supplementary Table 2) while the corresponding Faradic efficiency can be as high as 99%.”

Reviewer #2 (Remarks to the Author):

Wang et al. describes fabrication of P-doped core-shell NiMoO₄-hybrid zeolitic imidazolate framework for HER applications under large current. The authors need to address following points:

Response: We sincerely thank for the reviewer's comments and constructive suggestions which will greatly improve the depth and quality of our paper. We have carefully revised the manuscript (changes were marked and highlighted) and replied to the comments point-by-point shown below.

1. Why doping of P shows the beta NiMoO₄ phase?

Response: Thanks for the reviewer's comments. We have carefully revised our manuscript and added more profound and comprehensive discussions in both manuscript and supplementary materials about the metastable β -NiMoO₄ transformation mechanism and the role P plays during this process.

(1) We proposed the transformation mechanism of α -NiMoO₄ to β -NiMoO₄ based on the results of XPS, EPR spectra, and HAADF-STEM, etc. al. It is confirmed that P atoms exist in the form of PO₄³⁻ and take the place of Mo-O polyhedrons in the crystal structure of β -NiMoO₄ (PO₄³⁻ substitution), while there are some oxygen vacancies (V_o) in P-NiMoHZ sample (P- β -NiMoO₄). Herein, an additional description about the metastable β -NiMoO₄ transformation mechanism was added in the revised manuscript as follows:

Manuscript page 11: "At high temperature, α -NiMoO₄ would automatically transform into β -NiMoO₄ [*Inorg. Chem.* **7**, 1672-1675 (1968)]. Then, PO₄³⁻ enters into the lattice of NiMoO₄ and takes the place of M-O tetrahedron in the crystal structure of β -NiMoO₄ during phosphating (phosphate substitution, as shown in Fig. 2a), at the same time devoting to the thermodynamic optimization of the formation energy. The simultaneous carbonization process will provide preliminary V_o for charge balance. With the increase of the phosphating degree, the quantity of V_o also increases until the NiMoO₄ was completely stabilized at β phase."

Fig. 2 Crystal structures and electronic states of α -NiMoO₄ and P- β -NiMoO₄. a Schematic of crystal structure evolution after phosphate substitution.

(2) The effects of phosphate and oxygen vacancy for β phase stabilization was further discussed to support our proposed mechanism. The XRD results of a series of control groups (Supplementary Fig. 13) turn out that without phosphate substitution, β -NiMoO₄ would not be achieved (eg. C-NiMoHZ, $m = 0$ mg). However, from the EPR spectrum (Supplementary Fig. 15a), it has a certain amount of oxygen vacancies, which indicates a more significant role of PO_4^{3-} for beta phase stabilization.

Supplementary Figure 13. XRD patterns of P-NiMoHZ control groups prepared under certain conditions: (a) different amount of $\text{NaH}_2\text{PO}_2 \cdot \text{H}_2\text{O}$. (b) different reaction time.

Supplementary Figure 15. (a) EPR spectra of P-NiMoHZ control groups with different amount of $\text{NaH}_2\text{PO}_2 \cdot \text{H}_2\text{O}$ (left) and different reaction time (right). (g) Phosphorus atomic percent of these control groups.

Discussion added in the manuscript page 9: “From the EPR spectra and phosphorus atomic content comparison (Supplementary Fig. 15), the more phosphate groups could induce more oxygen vacancies. To this end, oxygen vacancies (positive charge) are deemed to balance the charge since PO_4^{3-} is more negative than MoO_4^{2-} when it occupies the place of Mo-O tetrahedrons.”

(3) To further prove the effects and the phosphate- V_o relationship mentioned above, we designed three more samples. Relevant discussions were also added in the revised manuscript as follows:

Manuscript page 10: “To confirm this hypothesis, three typical samples (CP-NiMoHZ, RP-NiMoHZ, and P-NiMoO₄, see details in experiment sections and Supplementary Fig. 2) were designed. The corresponding XRD patterns in Supplementary Fig. 17 demonstrate that α -NiMoO₄ could be converted into β phase by further phosphating reaction (C-NiMoHZ to CP-NiMoHZ; NiMoO₄ to P-NiMoO₄), verifying the indispensable role of phosphate. From EPR result, the existence of V_o in P-NiMoO₄ could also be substantiated (Supplementary Fig. 18). On the other hand, the partial transformation from β phase back to α phase (RP-NiMoHZ) after oxidative treatment highlights the subordinate function of V_o .”

Supplementary Figure 2. The synthetic routes for the various samples in this work and the corresponding abbreviation names.

Supplementary Figure 17. The XRD patterns of C-NiMoHZ, CP-NiMoHZ (Phosphating C-NiMoHZ), RP-NiMoHZ (calcining P-NiMoHZ in air atmosphere) and P-NiMoO₄ (Phosphating NiMoO₄).

Supplementary Figure 18. EPR spectrum of P-NiMoO₄.

2. Can you compare other non-metal doping such as nitrogen or others for HER application?

Response: Thanks for the reviewer’s suggestion. In order to carry out a more accurate comparison, we have further designed two non-metal (N and S) doping electrocatalysts by using the same NiMoO₄-hybrid zeolitic imidazolate framework (NiMoHZ) precursor. We have added the relevant comparisons and discussions in the manuscript and supplementary information.

(1) Synthesis description added in the experiment section (Manuscript page 24):

“**Synthesis of S-NiMoHZ and N-NiMoHZ.** The synthesis process was the same as P-NiMoHZ except for using CH₃C₆H₄SO₃H·H₂O and C₁₀H₁₆N₂O₈ as the sulfur source and nitrogen source, respectively.”

(2) Description added in the revised manuscript (Manuscript page 13):

“By comparison, the structure and the HER activity of the other two non-metal-doping electrocatalysts (N-NiMoHZ and S-NiMoHZ) were investigated (Supplementary Fig. 20-22).”

(3) Analysis added below the Supplementary Fig. 21 (SI page 25):

“According to the XRD patterns in supplementary Fig. 20d and 21d, the as-prepared N-NiMoHZ and S-NiMoHZ are mostly in the form of α -NiMoO₄. The small peaks marked by black star are assigned to ammonium molybdenum oxide and nickel sulfide, respectively, which only account for a tiny part of the materials. Integrating the XRD and the energy dispersive X-ray (EDX) mapping results, it could be confirmed that N and S have successfully incorporated into the materials.”

Supplementary Figure 20. (a) SEM image of N-NiMoHZ. Corresponding EDX elemental mapping of (b) Ni, (c) Mo, (e) O, and (f) N. (d) XRD pattern of N-NiMoHZ.

Supplementary Figure 21. (a) SEM image of S-NiMoHZ. Corresponding EDX elemental mapping of (b) O, (c) Ni, (e) Mo, and (f) S. (d) XRD pattern of S-NiMoHZ.

(4) Discussion added below the Supplementary Fig. 22 (SI page 26):

“The HER performance of N-NiMoHZ and S-NiMoHZ is much worse than that of P-NiMoHZ because of the unique structure of P- β -NiMoO₄ in P-NiMoHZ. Generally, heteroatom-doping would improve the HER activity of the target electrocatalysts [Chem. Soc. Rev. **49**, 1414-1448 (2020); Adv. Energy Mater. **10**, 1903870 (2020); Angew. Chem. Int. Ed. **60**, 14117–14123 (2021); ACS Energy Lett. **4**, 805-810 (2019)]. However, comparing these non-metal-doping samples, phosphate substituted β -NiMoO₄ plays a more important role in the enhancement of the HER performance, therefore displaying a much better HER performance.”

Supplementary Figure 22. Polarization curves of different non-metal-doping samples.

3. Generally, MOF (zeolitic imidazolate framework) suffers from low stability. However, P-doped core-shell NiMoO₄-hybrid zeolitic imidazolate framework (NiMoHZ) nanorod shows good stability. What are the reasons behind it?

Response: Thanks for the reviewer’s comment. We have to apologize that the abbreviated names in previous manuscript may have confused the reviewer about our optimal sample for HER. More detailed explanations have been added in the revised the manuscript accordingly.

(1) Indeed, MOF like zeolitic imidazolate framework usually suffer from two major drawbacks, poor electrochemical stability and low electrocatalytic activity [Sci. Adv. **7**, eabg2580 (2021); Chem. Commun. **54**, 7873 (2018); Chem **2**, 52 (2017)]. That is why in this work, we just used it as the intermediate precursor (NiMoO₄-hybrid zeolitic imidazolate framework, NiMoHZ) to prepare phosphate substituted β -NiMoO₄ (P- β -

NiMoO₄, the P-NiMoHZ sample, also mentioned in **Question 1**) as our optimal electrocatalyst for HER (as shown in Fig. 1a).

Fig. 1. Design and structure characterization of α -NiMoO₄ and β -NiMoO₄. a

Schematic illustration of the preparation process for P-NiMoHZ. The light green, light red, and yellow balls represent Ni²⁺ cations, MoO₄²⁻ anions, and NiMoO₄ crystals, respectively. The ball-and-stick models around nanorod arrays are vaporized 2-MIM (middle) and PH₃ (right) molecules.

2-MIM: 2-methylimidazole; PH₃: phosphine; NiMoHZ: NiMoO₄-hybrid zeolitic imidazolate framework; P-NiMoHZ: phosphated NiMoHZ.

(2) Since P- β -NiMoO₄ (the P-NiMoHZ sample) is the optimal electrocatalyst for HER and displays a superior stability at high current density, we would like to further analyze the reasons behind its high stability. More profound discussions about the origin of the high electrochemical stability were added in the revised manuscript as follows:

Manuscript page 16: “Basically, the high activity originates from the properties of materials and structures (eg. P- β -NiMoO₄ and fine-nanorod arrays), while the electrocatalytic stability comes from the material and structure stability. Therefore, the material and structure evolution are the key to unveil the origins of high electrocatalytic stability of P-NiMoHZ. Firstly, from the polarization curve of P-NiMoHZ in Fig. 3a, no redox peak is observed. It means that, during the HER electrocatalysis, P- β -NiMoO₄ is still the active material, and no extra reaction is taking place except for HER. The XRD patterns of P-NiMoHZ before and after stability test (Fig. 3e) also confirm this material stability. Secondly, SEM image after stability test (the inset of Fig. 3d) shows that the nano-morphology (structure) is well maintained, representing that the active site amount is not influenced.”

Fig. 3 Electrochemical properties for HER. **a** Polarization curves of NiMoO₄, C-NiMoHZ, P-NiMoHZ, P-NiMoO₄, P-NiMoHZ, and Pt/C in 1.0 M KOH saturated with N₂ gas at a scan rate of 5 mV s⁻¹. **b** Long-term stability tests of P-NiMoHZ at large current density (1000 mA cm⁻²). Inset: the SEM image of P-NiMoHZ after the stability test. **c** XRD patterns of the P-NiMoHZ before and after the stability test.

Manuscript page 16-17: “Furthermore, the bubble effects and hydrophilicity were further investigated to analyze the high stability. From the contact angle measurements (Supplementary Movie 1-2 for P-NiMoHZ sample, Supplementary Fig. 29), in contrast to NiMoO₄, P-NiMoHZ is hydrophilic and much more gas-phobic. Therefore, from macroscopic view, during HER process, the produced H₂ gas bubble will be easily released on the electrode surface. At the same time, the surface will quickly get wet by water again. Thus, at the same current density, the size of bubbles on the P-NiMoHZ electrode surface would be far smaller than that on the NiMoO₄ electrode (Fig. 3f), which was confirmed by the bubble size distribution on P-NiMoHZ and NiMoO₄ during HER electrocatalysis (Supplementary Fig. 30). As a result, it will avoid the catalyst shedding issue from large drastic bubble releasing and facilitate the retention of the nanostructure on the electrode surface. Meanwhile, the faster releasing of the bubbles would lead to faster re-exposure of the active sites.”

Supplementary Figure 29. Contact angle measurements for NiMoO₄ sample. (a) water contact angle, (b) H₂ contact angle. The water contact angle is 102°. The H₂ contact angle is 127°. These results indicate that NiMoO₄ is hydrophobic and more gas-philic than P-NiMoHZ.

Fig. 3 **Electrocatalytic properties for HER.** f Photographs of the electrode during HER electrocatalysis (left: NiMoO₄; right: P-NiMoHZ).

Supplementary Figure 30. Bubble size distributions on NiMoO₄ (a) and P-NiMoHZ (b) electrodes. In total forty bubbles were considered on each electrode.

Bubble size distribution on both electrodes was recorded according to the following instructions and added in the revised manuscript:

Manuscript page 25: “The bubble size distributions were observed under chronopotentiometry mode at the current density of 100 mA cm⁻².”

Reviewer #3 (Remarks to the Author):

This study reports highly organized, high surface area electrode based on Ni-Mo for hydrogen evolution reaction. Uniqueness of the study is to synthesize Ni-Mo-O species via sublimation-vapor phase transformation (SVPT) strategy that the authors develop. This leads to the regular rod-shaped arrays on the Ni foam substrate. The obtained electrode shows high current with low overpotential with good stability for HER, even at high current density region. The combination of Ni-Mo has already been applied for long time in alkaline electrolyzer, and there is boosting fashion of reporting this material in recent years. Such recent activities include Angew. Chem. Int. Ed. 2021, 60, 7051, Angew. Chem. Int. Ed. 2021, 60, 5771. Basically, what is missing in this article is correct description of how the experiments were conducted, and how the calculations were introduced to identify which properties of the materials. To my view, the paper is poorly written, and it is not ready for publication. NiMo oxide should be (at least partly) reduced upon HER condition. Scientific content, if any, should be well improved with recent understanding of this particular material.

Response: We sincerely thank the comment from the reviewer. The suggestions proposed are valuable and helpful for improving this paper. We have carefully revised the manuscript (changes were marked and highlighted) and replied to the comments point-by-point shown below.

(1) Indeed, combination of Ni-Mo has already been applied for HER electrocatalysis in alkaline for a long time. However, in fact as shown in Table R1, these electrocatalysts are the form of NiMo alloy, NiMo alloy/oxide composite, NiMo sulfide, NiMo nitride, or the other NiMo composite materials.

Table R1. The reference of different NiMo-based catalysts in HER

NiMo-based electrocatalysts types	References
NiMo alloy	Angew. Chem. Int. Ed. 60 , 7051-7055 (2021)
	Angew. Chem. Int. Ed. 60 , 5771-5777 (2021)
	Adv. Energy Mater. 11 , 2003511 (2021)
	Adv. Funct. Mater. 29 , 1903747 (2019)
NiMo alloy/oxide composite	Adv. Energy Mater. 9 , 1901454 (2019)
	Adv. Mater. 29 , 1703311 (2017)

	Nat. Commun. 8 , 15437 (2017)
NiMo sulfide	Appl. Catal. B Environ. 276 , 119137 (2020)
	ACS Catal. 7 , 6179–6187 (2017)
NiMo nitride	Nano Energy 78 , 105375 (2020)
	Nat. Commun. 10 , 5106 (2019)
NiMo composite materials	ACS Energy Lett. 4 , 3002–3010 (2019)
	Adv. Mater. 30 , 1803151 (2018)
	ACS Catal. 8 , 8107–8114 (2018)

(2) **NiMo oxide and other non-noble transition metal oxides** directly as the active materials for HER have not been paid much attention to because of their inappropriate hydrogen adsorption strength and low conductivity [*Chem. Soc. Rev.* **49**, 9154–9196 (2020)], although they are cost-effective, rich in reserves, and easy to be prepared. Some previously reported works has already made some progress by heteroatom-doping strategy [*Angew. Chem. Int. Ed.* **60**, 14117–14123 (2021); *Adv. Funct. Mater.* **29**, 1805298 (2019)], but their activity and performance are still not good enough for potential industrial applications. To this end, in this work, we put forward a novel strategy to prepare **phosphate substituted NiMoO₄ in metastable β phase**, which exhibits an outstanding HER activity at both small and large current density. Meanwhile, the active electronic states are introduced after the phosphate substitution and phase transformation, which are critical to the enhanced HER activity. Furthermore, we found that it shows a stable performance at 1000 mA cm⁻² during long-term test, which might meet the industrial application requirements.

(3) Regarding the synthesis and electrochemical measurements sections, some vague descriptions might cause misunderstanding to readers. We have carefully revised our manuscript and provided a more detailed one in the Methods section (also see in the response of **Question 5** and **Question 7**).

(4) For the calculation information, both thermodynamics and kinetics calculation, it is identified that the P- β -NiMoO₄ system as the high-efficiency HER catalyst in alkaline media not only affords neutral hydrogen source (Ni site), but also has appropriate hydrogen binding energy (O1 site) for HER activity. Especially, from the kinetic viewpoints, it is performed climbing image nudge elastic band calculations [*J. Chem.*

Phys. 140, 214106 (2014)] on each of these combinations of the final H–OH configuration with the most stable initial H₂O configuration and selected the combination with the least energy barrier. In addition, from the view of thermodynamics, the hydrogen adsorption free energy ΔG_{H^*} is an effective descriptor to determine the HER activity. The optimal value of ΔG_{H^*} is close to 0.0 eV, where adsorbed atomic hydrogen is in a thermoneutral state and can perform efficient proton/electron-transferred and hydrogen release [*J. Am. Chem. Soc.* **127**, 5308 (2005)].

(5) We admit that our previous version of the manuscript was not organized in appropriate and comfortable logic. We have **revised the whole manuscript** by starting with the material characterization and phase transformation mechanism, then the prediction of HER activity from active electronic states, subsequently the comprehensive electrochemical measurements of the as-prepared materials and the investigation on the origin of the increased electrocatalytic stability, finally the theoretical calculations to unveil the active sites of our phosphate substituted β -NiMoO₄ (P- β -NiMoO₄) (also see in the response of **Question 2**). Meanwhile, we have corrected the inaccurate narratives in the manuscript (also see in the response of **Question 1**).

(6) Regarding the NiMo oxide reduction upon HER condition, we have added additional discussions in the revised manuscript:

Manuscript page 16: “From the polarization curve of P-NiMoHZ in Fig. 3a, no redox peak is observed. This means that, during the HER electrocatalysis, P- β -NiMoO₄ is still the active material, and no extra reaction is taking place except for HER. The XRD patterns of P-NiMoHZ before and after stability test (Fig. 3e) also confirm the stability of this material.”

Fig. 3 Electrochemical properties for HER. **a** Polarization curves of NiMoO₄, C-NiMoHZ, P-NiMoO₄, P-NiMoHZ, and Pt/C in 1.0 M KOH saturated with N₂ gas at a scan rate of 5 mV s⁻¹. **e** XRD patterns of the P-NiMoHZ before and after the stability test.

Therefore, based on the above analysis, NiMo oxide is served as the active catalyst and keeps stable during the HER electrocatalysis, indicating that it would not be reduced upon HER condition in this work.

(7) Since the mechanisms in recently published works are ascribed to NiMo alloy, we further used theoretical calculations to unveil the active sites and electrocatalytic mechanism for our P-β-NiMoO₄. For HER in alkaline conditions, there are two continuous steps: water dissociation and hydrogen desorption. Based on the thermodynamics and kinetics calculation, it is proved that the active Ni site of P-β-NiMoO₄ system preferred to decompose H₂O to supply the neutral hydrogen source, and the active O site connecting P atom has appropriate hydrogen binding energy in favor of hydrogen production (added H₂O dissociation calculation see in the response of **Question 9**).

1. First, general scientific narrative should be improved, which reflects the quality of the science of the authors. For example, Abstract starts with “Transition Metal oxides (TMOs) are abundant in nature”, where noble metal oxides including Pt and Ir oxides are also TMOs.

Response: We sincerely thank the reviewer’s comment. We admit that in our previous manuscript some narratives were inaccurate. We have carefully revised them and highlighted the changes in the manuscript.

(1) Revision in the abstract: “Non-noble transition metal oxides are abundant in nature. However, they are widely regarded as catalytically inert for hydrogen evolution reaction (HER) due to their scarce active electronic states near the Fermi-level. How to largely improve the HER activity of these kinds of materials remains a great challenge. Herein, as a proof-of-concept, we use a non-solvent strategy to achieve phosphate substitution and the subsequent metastable crystal phase stabilization of NiMoO₄, which could

efficiently generate the active electronic states of NiMoO₄ and promote its intrinsic HER activity. Phosphate substitution is proved to be imperative for the stabilization and activation of β-NiMoO₄, which exhibits the optimal hydrogen adsorption free energy (-0.046 eV) and ultralow overpotential of -23 mV at 100 mA cm⁻² in 1 M KOH for HER. Especially, it also maintained long-term stability for 200 h at the large current density of 1000 mA cm⁻² with an overpotential of -210 mV. This work provides a new route for activating transition metal oxides for HER by stabilization of metastable phases with abundant active electronic states.”

(2) Revision in Discussion: “In summary, we developed a facile non-solvent strategy to manipulate the active electronic states of α-NiMoO₄ through phosphate substitution and subsequent phase transformation. Abundant active electronic states in the metastable phase P-β-NiMoO₄ (P-NiMoHZ) were achieved and greatly promote its intrinsic HER activity in alkaline. The phase transformation mechanism unveils that phosphate substitution and oxygen vacancy play a synergistic role in the stabilization of the metastable phase, respectively. Furthermore, we highlight the significant effects of the hydrophilic and gas-phobic ability of the electrocatalysts on their stability and kinetics at large current density. This study paves a new avenue for designing low-cost and highly active non-noble transition metal oxides, especially for HER applications under large current density.”

2. The paper is written in an unintellectual order of messages. Results start with density of state calculation, before even talking about how “P-doping” is achieved and where the location of P species is located. How can the readers be convinced with the calculation? Even if it is accepted, what is the evidence that PO₄ substitution makes metallic character (half filled) of Ni? What is the size of the cell to obtain this?

Response: We sincerely thank the reviewer’s valuable comment. We agree with the reviewer’s viewpoint that the writing logic of our previous manuscript was not very proper, which might cause doubts to readers about the feasibility of our theoretical calculations on electronic state.

We have reorganized the writing sequence and put the electronic state calculation part after the material characterizations and phase transformation mechanism part. Figure 1 in the previous version of the manuscript now has been switched to Figure 2. The models for calculation were created based on the characterization/experiment results that phosphate (PO_4^{3-}) takes the place of Mo-O tetrahedron in the crystal structure of β -NiMoO₄. The revision parts were highlighted in the manuscript.

Revision in the manuscript page 11: “**Electronic state analysis of NiMoO₄.**”

(1) We have added characterization discussion to support the theoretical calculation results in the revise manuscript page 11:

As shown in Fig. 2b, after the phosphate treatment, there is a slight shift to higher binding energy of the characteristic Ni 2p_{1/2} and 2p_{3/2} peaks in P- β -NiMoO₄, which imply that the partial charge of Ni atom after phosphate treatment are transferred to neighboring atoms such as P and O atom.

Fig. 2 Crystal structures and electronic states of α -NiMoO₄ and P- β -NiMoO₄. a Schematic of crystal structure evolution after phosphate substitution. **b** XPS spectra of

Ni 2p in P- β -NiMoO₄ and α -NiMoO₄. c Schematic illustration of the active electric states in different phase NiMoO₄.

(2) Theoretical calculations were then applied to predict the HER electrocatalytic activity of α -NiMoO₄ and P- β -NiMoO₄ (phosphate substituted β -NiMoO₄). There are some optimizing electronic states originated from Ni atom near the Fermi level in P- β -NiMoO₄ compared to the α -NiMoO₄ system, which facilitates the charge transfer from activated Ni to surrounding atoms (Fig. 2c). Similar to that of previous work, the generation of active electron states are mainly attributed to the uplifting the states of Ni-3d in P- β -NiMoO₄ system. Thus, the adsorbed protons could easily receive the electrons to produce hydrogen atoms, thus accelerating the whole HER process on P- β -NiMoO₄.”

(3) Regarding the size of the cell, the computational model supercell (P- β -NiMoO₄-(110)) containing 23 atoms was introduced to model a system where one Mo atom is substituted by a P atom, approaching the isolated impurity limit. The modulus unit cell vector in the z direction was set to 15 Å, which led to negligible interactions between the system and their mirror images.

(4) As for the location of phosphate and how phosphate substitution is achieved, they have been thoroughly explained in the responses to **Question 3 (discussion on locations)** and **Question 5 (description of experiments)**.

3. For this Figure 1, after reading carefully the latter part of study, P seems to be present as phosphate, which is reasonable, but what was the evidence to substitute MoO₄ site? Just having the same structure is not sufficient.

Response: We sincerely thank the reviewer’s comment. We have carefully revised the manuscript to give a more comprehensive discussion regarding the phosphate substitution site. The revision parts were highlighted by yellow highlight in the manuscript.

(1) Based on the XPS P 2p spectra, it is clear that P exists in the form of phosphate in P-NiMoHZ, which was also agreed by the reviewer. From the line scans profile (Supplementary Fig. 12b), the same signal positions of P, Ni, Mo, and O suggest that P enter into the lattice of β -NiMoO₄ rather than the outer carbon layer.

Supplementary Figure 12. (b) Cross-sectional elemental distributions by line scans along the white line.

(2) The atomic arrangement of NiMoO_4 matches well with the theoretical (101) plane of the α - NiMoO_4 unit cell (Supplementary Fig. 9). The periodical distributed atoms of P-NiMoHZ (Fig. 1e) are consistent with the (110) plane of the β - NiMoO_4 unit cell model. When using these theoretical crystal models to simulate XRD patterns, the obtained peaks match well with the actual experimental ones (Supplementary Fig. 10), further indicating the practical structures are in high conformity with the crystal models. Therefore, we can assure that the bright dots in the HAADF-STEM represent the metal atoms.

Supplementary Figure 9. (a) (101) plane model of α - NiMoO_4 . The blue, green, and red balls represent Mo, Ni, and O atoms, respectively, while black, blue, and red lines represent cell axes. (b) High-resolution HAADF-STEM image of NiMoO_4 synthesized by dehydration of $\text{NiMoO}_4 \cdot x\text{H}_2\text{O}$. The bright dots represent metal atoms.

Fig. 1 Design and structure characterization of α -NiMoO₄ and β -NiMoO₄. e HAADF-STEM image of P-NiMoHZ. The blue and green circles represent Mo and Ni atoms, respectively.

Supplementary Figure 10. Unit cell schematics of (a) α -NiMoO₄. (b) β -NiMoO₄, the blue, green, and red balls represent Mo, Ni, and O atoms, respectively. (c) Comparisons between simulated theoretical characteristic peaks and XRD patterns of NiMoO₄ and P-NiMoHZ.

(3) From the amplified HAADF-STEM image of P-NiMoHZ (Supplementary Fig. 11), in addition to the regular configuration, some darkened sites could be observed, which is due to the loss of the original atoms in those sites. Since the bright dots represent the metal atoms, according to the analysis of XPS, the relatively dark site marked by the red arrow might be the substitution of a Mo atom by a P atom, which results from the replacing of Mo-O tetrahedron by PO₄³⁻ tetrahedron. It could be approved by the intensity profile of HAADF-STEM as well. As shown in Fig. 1f, when P atoms occupy the sites of metal atoms, the intensity undergoes a sudden decrease due to the smaller atomic size of phosphorus compared to nickel and molybdenum.

Supplementary Figure 11. Amplified HAADF-STEM image of P-NiMoHZ. V_o are marked by orange arrows, while P substitution by the red arrow.

Fig. 1 Design and structure characterization of α -NiMoO₄ and β -NiMoO₄. f HAADF-STEM image of P-NiMoHZ with an intensity profile corresponding to the red line.

(4) Except for the material characterization, we further added an additional metal vacancy formation energies calculation to prove our findings. Description added in the manuscript: “Theoretical calculations of the metal vacancy formation energy in β -NiMoO₄ (Supplementary Fig. 16) suggest that the Mo vacancy compound model is thermodynamically more stable than Ni vacancy, which thus is more favorable for the substitution of phosphate.”

Supplementary Figure 16. (a) The formation energies of proposed configurations of Ni and Mo vacancy in β -NiMoO₄ system. The formation energies are calculated as $E_f = E(\beta\text{-NiMoO}_4\text{-}\square_x) + xE(\text{TM}) - E(\beta\text{-NiMoO}_4)$ (TM= Ni or Mo, \square = Ni or Mo vacancy, x is the number of TM vacancy per unit cell).

4. Also I recommend the appropriate use of P doping and phosphate substitution. P doping is typically used for semiconductor (such as Si). This is P substitution as Fig. 1 caption says, but more specifically phosphate substitution.

Response: We sincerely thank the reviewer's comment. In fact, according to our characterizations, it should be the phosphate substitution instead of P doping in this work. We have already revised and changed them to phosphate substitution. Besides, we added a schematic illustration of NiMoO₄ crystal evolution after phosphate substitution to elucidate the structure changes (Fig. 2a).

Fig. 2 Crystal structures and electronic states of α -NiMoO₄ and P- β -NiMoO₄. a Schematic of crystal structure evolution during phosphate substitution.

5. Related to above phosphate substitution, it is unclear how the experiment is conducted. Solid hypophosphite is mixed in crucible? How reproducible is this experiment?

Response: We sincerely thank the reviewer's comment. We are sorry that our previous description was not detailed enough to explain the process. We have carefully revised our manuscript (changes were marked by yellow highlight).

(1) As shown in Supplementary Fig. 1b, for Sublimation-Vapor Phase transformation (SVPT), the substrate is kept away from the solid powder reactant by putting it on the hollow quartz tube. Similarly, for the phosphating process, solid sodium hypophosphite ($\text{NaH}_2\text{PO}_2 \cdot \text{H}_2\text{O}$) was put at the bottom of a ceramic crucible as well, while the substrate was put on the quartz tube inside the ceramic crucible with a lid.

Supplementary Figure 1. (b) Photograph of the actual reaction device.

(2) Revision in the experiment section: “**Synthesis of P-NiMoHZ on Ni foam.** In a typical process, 90 mg solid $\text{NaH}_2\text{PO}_2 \cdot \text{H}_2\text{O}$ was put at the bottom of a ceramic crucible, while NiMoHZ ($1.0 \times 1.6 \text{ cm}^2$) was kept away from the solid powder reactant by vertically putting it on a hollow quartz tube inside the crucible. Afterwards, the crucible was put in a tube furnace at $500 \text{ }^\circ\text{C}$ for 60 min (Ar atmosphere) to get P-NiMoHZ (90 mg, $500 \text{ }^\circ\text{C}$, 60 min). During the heating process, $\text{NaH}_2\text{PO}_2 \cdot \text{H}_2\text{O}$ would decompose and produce PH_3 gas by the following equation:

While keeping the other parameters, the other eight control groups were obtained by only changing the reaction temperature ($400 \text{ }^\circ\text{C}$, $450 \text{ }^\circ\text{C}$, and $550 \text{ }^\circ\text{C}$), the feed amount of $\text{NaH}_2\text{PO}_2 \cdot \text{H}_2\text{O}$ (30 mg, 60 mg, and 120 mg), or the reaction time (30 min and 90 min), and denoted as P- $400 \text{ }^\circ\text{C}$, P-30 mg, P-30 min, and etc.”

(3) PH_3 reacts with NiMoO_4 to give PO_4^{3-} and oxygen vacancies. However, as shown in Supplementary Fig. 7, excessive temperature will lead to over-phosphating and completely convert precursor into nickel phosphide and nickel phosphate.

Supplementary Figure 7. (a) XRD pattern of P-NiMoHZ obtained at 550 °C; (b) Corresponding SEM image.

Analysis added below the Supplementary Fig. 7: “The XRD pattern obtained at 550 °C shows that excessive temperature would greatly increase the phosphating degree, making the precursor completely be converted into the composite of Ni₃P, Ni₃(PO₄)₂, and MoO₂ instead of β-NiMoO₄. The process follows by the below reaction equations:

(4) As to the reproducibility of the experiment, the synthesis process is quite convenient. It is easy to follow our protocols to prepare the electrocatalysts. We have also prepared the samples several times, the corresponding XRD patterns are as followed.

Figure R2. The XRD patterns of the P-NiMoHZ with several samples.

6. Recent understanding of Ni-Mo species for HER is that the electrode is “alive” upon the potential sweeping: the dissolution and reorganization of Mo species on the surface happens. While NiMo alloy is commonly being accepted NOT to be the active site, but more and more papers discuss the crucial role of Mo oxides on the surface as the true active redox species. See, e.g., ACS Catal. 2020, 10, 12858.

Response: We sincerely thank the reviewer's comment and appreciate the reviewer's professionalism.

(1) The recently reported findings in experiments and theoretical calculations elucidate a novel process regarding the dissolution and reorganization of Mo species [*Angew. Chem. Int. Ed.* **60**, 7051–7055 (2021); *ACS Catal.* **10**, 12858-2866 (2020)]. However, in fact, these mechanisms are referred to the HER electrocatalysts of NiMo alloy, like MoNi₄ (Figure R3). In this work, we put forward a novel strategy to prepare **phosphate substituted NiMoO₄ in metastable β phase (P- β -NiMoO₄)** which exhibits an outstanding HER activity at both small and large current density. The active sites of our P- β -NiMoO₄ are quite different from those of NiMo alloys.

Figure R3. Selected area electron diffraction pattern of Ni_xMo_y catalyst and converted spectrum. [*ACS Catal.* **10**, 12858-12866 (2020)]

(2) Discussion added in the activity and stability at large current density section: “From the polarization curve of P-NiMoHZ in Fig. 3a, no redox peak is observed. This means that, during the HER electrocatalysis, P- β -NiMoO₄ is still the active material, and no extra reaction is taking place except for HER. The XRD patterns of P-NiMoHZ before and after stability test (Fig. 3e) also confirm the stability of this material.” Therefore, based on the above analysis, NiMo oxide is served as the active catalyst and keeps stable during the HER electrocatalysis, indicating that it would not be reduced upon HER condition in our work. No extra NiMo alloy was observed from our characterizations and experiments.

Fig. 3 Electrochemical properties for HER. **a** Polarization curves of NiMoO₄, C-NiMoHZ, P-NiMoO₄, P-NiMoHZ, and Pt/C in 1.0 M KOH saturated with N₂ gas at a scan rate of 5 mV s⁻¹. **e** XRD patterns of the P-NiMoHZ before and after the stability test.

(3) Since P-β-NiMoO₄ is in the oxide phase, the mechanisms for HER would be largely different from those of NiMo alloys. Therefore, we further use theoretical calculations to unveil the active sites and electrocatalytic mechanism for our P-β-NiMoO₄. For HER in alkaline conditions, there are two continuous steps: water dissociation and hydrogen desorption. Based on the thermodynamics and kinetics calculation, it is proved that the active Ni site of P-β-NiMoO₄ system preferred to decompose H₂O to supply the neutral hydrogen source, and the active O site connecting P atom has appropriate hydrogen binding energy in favor of hydrogen production (added H₂O dissociation calculation see in the response of **Question 9**).

7. Current density of > 1000 mA cm⁻² is reported. This type of measurement requires special care of cell design as the solution resistance and counter electrode should be adequately optimized. The description of experiment is not sufficient. What is the size of working electrode? At counter electrode, what is the size and how to catch up the 1A with carbon electrode (self-oxidation)? It is also questionable how iR correction is achieved if any. If iR correction is not achieved, obtained result is highly skeptical as it should reflect the iR loss in the measured potential. If iR correction is done, slight difference in this value would change the description of the performance. There should be bubble problem of the H₂ product. What is the power source used to achieve this high current range?

Response: We sincerely thank the reviewer's question. We have answered the questions accordingly and revised the manuscript (changes were marked and highlighted).

(1) The set-up device for stability test was referred to normally used three-electrode system. The parameter setting was also similar. [*Nat. Mater.* **18**, 1309–1314 (2019); *Nat. Commun.* **10**, 269 (2019); *Nat. Commun.* **9**, 2551 (2018)]

(2) Revised electrochemical measurements section: “The as-prepared catalysts were directly applied as the working electrode, while saturated calomel electrode (SCE) and polished graphite rod were used as the reference electrode and the counter electrode, respectively. The working electrode area immersed in the electrolyte was $0.5 \times 0.5 \text{ cm}^2$.”

(3) According to the previously reported work, [*Nat. Mater.* **18**, 1309–1314 (2019)] the type of counter electrode will not influence the electrocatalytic performance at large current density, as shown in Figure R4. Therefore, we just used the typical polished graphite rod as the counter electrode.

Figure R4. The effects on polarization curve of different counter electrode type. [*Nat. Mater.* **18**, 1309–1314 (2019)]

(4) The revised experiment description regarding the stability test at 1000 mA cm^{-2} : “The stability tests were conducted by chronopotentiometry at the current density of 10 mA cm^{-2} for 10 h and 1000 mA cm^{-2} for 200 h, respectively.” Since we adopted chronopotentiometry mode to test the stability, then the current density applied on the electrode would keep at a constant (1000 mA cm^{-2}) during the test. Therefore, the applied current is 250 mA. Due to the existence of the solution resistance, then the overpotential required would be larger than the case with iR compensation. This is consistent with the stability test result. The overpotential is about 225 mV at 1000 mA cm^{-2} (Fig. 3d) compared to about 200 mV at the same current density in the polarization curve with iR compensation (Fig. 3a). This result (225 mV at 1000 mA cm^{-2}) has already

been comparable to or even better than many reported electrocatalysts. [*J. Am. Chem. Soc.* **143**, 8720–8730 (2021); *Nat. Commun.* **10**, 269 (2019); *Nat. Commun.* **9**, 2551 (2018)]

Fig. 3 Electrochemical properties for HER. **a** Polarization curves of NiMoO₄, C-NiMoHZ, P-NiMoO₄, P-NiMoHZ, and Pt/C in 1.0 M KOH saturated with N₂ gas at a scan rate of 5 mV s⁻¹. **d** Long-term stability tests of P-NiMoHZ at large current density (1000 mA cm⁻²). Inset: the SEM image of P-NiMoHZ after the stability test.

(5) Bubble issues might be a huge challenge to the overall performance and stability of the electrocatalysts. Because the unreleased H₂ bubble would make the active sites within the bubble area cannot reach to the reactant, H₂O, in alkaline, thus decreasing the actual working active site number. On the other hand, over high releasing speed of the bubbles might cause the shedding of the electrocatalysts on the electrode surface. To further study the bubble effects, we conducted the contact angle measurements and recorded the bubble size distribution on electrodes. Discussion added in the manuscript: “Furthermore, the bubble effects and hydrophilicity were further investigated to analyze the high stability. From the contact angle measurements (Supplementary Movie 1-2 for P-NiMoHZ sample, Supplementary Fig. 29), it demonstrates that, in contrast to NiMoO₄, P-NiMoHZ is hydrophilic and much more gas-phobic. Therefore, from macroscopic view, during HER process the produced H₂ gas bubble will be easily released on the electrode surface, at the same time the surface will quickly get wet by water again. The overall effect is that at the same current density the size of bubbles on the P-NiMoHZ electrode surface would be much smaller than those on the NiMoO₄ electrode (Fig. 3f), which was confirmed by the bubble size distribution on P-NiMoHZ and NiMoO₄ during HER electrocatalysis (Supplementary Fig. 30). As a result, this phenomenon will avoid the catalyst shedding issue from drastic large bubble releasing

and facilitate the retention of the nanostructure on the electrode surface. Meanwhile, the faster releasing of the bubbles would lead to faster re-exposure of the active sites, which in favor of electrocatalysis.” Bubble size distribution on both electrodes was recorded according to the following information added in the electrochemical measurements section: “The bubble size distributions were observed under chronopotentiometry mode at the current density of 100 mA cm^{-2} .”

Supplementary Figure 29. Contact angle measurements for NiMoO₄ sample. (a) water contact angle, (b) H₂ contact angle. The water contact angle is 102°. The H₂ contact angle is 127°. These results indicate that NiMoO₄ is hydrophobic and more gas-philic than P-NiMoHZ.

Fig. 3 Electrocatalytic properties for HER. f Photographs of the electrode during HER electrocatalysis (left: NiMoO₄; right: P-NiMoHZ).

Supplementary Figure 30. Bubble size distributions on NiMoO₄ (a) and P-NiMoHZ

(b) electrodes. In total forty bubbles were considered on each electrode.

8. For capacitance correlation to calculate TOF, it is true that many papers discuss this, but my personal view is not essential. Highly porous, high surface area material, ECSA would underestimate the active surface. This material shows high current density per geometric area, so that the

Response: Thanks for the reviewer's comment. In this work, we use a general ECSA strategy to calculate the per-site turnover frequency for intrinsic activity evaluation [*Energy Environ. Sci.* **8**, 3022-3029 (2015)].

We agree that for highly porous, high surface area material, this method might overestimate the value of TOF to some extent. Since our electrocatalyst displays an excellent overall HER performance, it must possess high intrinsic activity, a large number of active sites, good conductivity, and fast mass transfer.

Therefore, we just put this part into the supplementary information for reference and tabulate a comparison table.

Supplementary Table 3. The comparison of TOF values at 100 mV overpotential of different electrocatalysts

Electrocatalyst	Electrolyte	TOF	Reference
P-NiMoHZ	1 M KOH	0.76	This work
NiCo ₂ P _x	1 M KOH	0.056	Adv. Mater. , 29 , 1605502, (2017)
Ni-MoS ₂	1 M KOH	0.08	Energy Environ. Sci. , 9 , 2789, (2016)
Ni ₅ P ₄	1 M KOH	0.06	Energy Environ. Sci. 8 , 1027–1034, (2015)
FeB ₂	1 M KOH	0.165	Adv. Energy Mater. 7 , 1700513, (2017)
GDY/MoO ₃	0.1 M KOH	0.22	J. Am. Chem. Soc. 143 , 8720–8730, (2021)
NiCo ₂ P _x	1 M KOH	0.056	Adv. Mater. , 29 , 1605502, (2017)
S-CoO NRs	1 M KOH	0.41	Nat. Commun. 8 , 1509, (2017)
CoN _x /C	1 M KOH	0.39	Nat. Commun. 6 , 7992, (2015)

9. Towards the latter part, the discussion starts to talk about H₂O adsorption etc. But it is commonly considered that the alkaline HER is pinned by the rate controlling O-H bond dissociation of H₂O. Although the thermodynamic calculation discussed in this

study is commonly done, it is irrelevant for alkaline HER. For example, H_2O adsorption is not the rate determining step at all.

Response: We sincerely appreciate the reviewer's professional comment.

(1) Following the reviewer's proposals, we consider firstly the dissociation process of H_2O in alkaline media. According to previous reports [Angew. Chem. Int. Ed. **57**, 7568, 2018, Adv. Sci. **8**, 2001881, (2021), Angew. Chem. Int. Ed. **58**, 12014–12017, (2019)], the barrier height of water dissociation also plays an important role in determining overall alkaline HER reaction kinetic rate. In Supplementary Fig. 32a and Figure 4c, for the water dissociation, it is found that the active Ni sites of P- β -NiMoO₄ (b) system exhibit much lower activation barriers (0.569 eV) than that of α -NiMoO₄ (1.457 eV) and Pt catalysis (0.94 eV) [J. Am. Chem. Soc. **138**, 161174 (2016)]. From the kinetic viewpoints, the P- β -NiMoO₄ system could accelerate water dissociation to provide neutral hydrogen source. Both thermodynamics and kinetics calculation, it is proved that the P- β -NiMoO₄ system as the high-efficiency HER catalyst in alkaline media not only affords neutral hydrogen source, but also has appropriate hydrogen binding energy.

Supplementary Figure 33. Water dissociation barrier for reaction pathway of α -NiMoO₄ system. The insets are the structure of the corresponding IS (initial state), TS (transition state) and FS (final state). The colors of elements are: green for Ni, blue for Mo, red for O and white for H.

Fig. 4 Calculated adsorption free energy and electronic structure of different adsorbates between α -NiMoO₄-(110) and P- β -NiMoO₄-(110) surface. **c** Water dissociation barrier for reaction pathway of P- β -NiMoO₄ system. The insets are the structure of the corresponding IS (initial state), TS (transition state) and FS (final state).

(2) We have added the relate discussion and analysis in the revised manuscript:

Manuscript page 19: **Theoretical simulation.** To understand the origin of the high catalytic activity, DFT calculations were performed to investigate a cooperative catalytic mechanism of P- β -NiMoO₄ (P-NiMoHZ). For HER in alkaline conditions, there are two continuous steps: water dissociation and hydrogen desorption (Angew. Chem. Int. Ed. 2018, 57, 7568, Adv. Sci. 2021, 8, 2001881, *Nat. Commun.* **10**, 149, (2021)). Especially, the considerably slower rate of the water dissociation in the alkaline electrolyte has greatly hindered the overall high purity hydrogen production and reaction kinetic rate. From the kinetic viewpoints, the lower activation barriers of water dissociation of H₂O plays an important role in accelerating to provide a neutral hydrogen source. The calculation results reveal that the active Ni sites of P- β -NiMoO₄ (Fig. 4a, b) exhibit much higher adsorption energy (-0.599 eV) for water dissociation, compared to α -NiMoO₄ (-0.424 eV) (Supplementary Fig. 31). In addition, it is found that there is a linear correlation among H₂O adsorption energy, bond length and the amount of charge transfer Δe . The more active the electron transfer, the stronger the H₂O binding energy, and the shorter the α -NiMoO₄/P- β -NiMoO₄-H₂O bond length (Supplementary Fig. 32). Herein, higher H₂O adsorption energy corresponds to the larger amount of charge transfer benefit from lowering the activation barrier of water dissociation. In Fig. 4c, it is found that the active Ni sites of P- β -NiMoO₄ (b) system exhibit much lower activation barriers (0.569 eV) than that of α -NiMoO₄ (1.457 eV)

and Pt catalysis (0.94 eV) (Supplementary Fig. 33). [J. Am. Chem. Soc. **138**, 161174 (2016)]. From the kinetic viewpoints, the P- β -NiMoO₄ system could accelerate water dissociation to provide neutral hydrogen source. In Fig. 4d, the corresponding charge density differences of H₂O adsorbed on Ni site in P- β -NiMoO₄ are also represented, qualitatively reflecting the redistribution of electron states with the largest amount of charge transfer, further optimizing the decomposition of water.

Besides the H₂O dissociation barrier, the hydrogen adsorption free energy ΔG_{H^*} is also an effective descriptor to estimate the HER activity. Herein, different exposed atomic sites of P- β -NiMoO₄ and α -NiMoO₄ were used to calculate ΔG_{H^*} (Fig. 4e, Supplementary Fig. 34).

Therefore, compared to α -NiMoO₄, the P- β -NiMoO₄ system prefers to regulate the intrinsic charge distribution of exposed atoms, further optimizing the HER performance.

Fig. 4 Calculated adsorption free energy and electronic structure of different adsorbates between α -NiMoO₄(110) and P- β -NiMoO₄(110) surface. **a** The optimized structure of P- β -NiMoO₄(110). Especially, the Mo atom was substituted by the P atom according to the experiment verification. **b** The Gibbs free energy diagram

on H₂O adsorbed on different sites in α -NiMoO₄-(110) and P- β -NiMoO₄-(110) surface, respectively. **c** Water dissociation barrier for reaction pathway of P- β -NiMoO₄ system. The insets are the structure of the corresponding IS (initial state), TS (transition state) and FS (final state). **d** The corresponding charge density differences of H₂O adsorbed on Ni sites in P- β -NiMoO₄. The yellow and blue regions indicate the accumulated or dispersed amount of electron states of atoms around the interface, respectively. **e** Hydrogen adsorption free energy (ΔG_{H^*}) in different exposed atoms in P- β -NiMoO₄-(110). Here, these exposed atoms are considered as the catalytic sites for HER. **f** Linear correlation between ΔG_{H^*} , P- β -NiMoO₄-H bond length, α -NiMoO₄-H bond length and the amount of charge transfer Δe of various active sites in different phase of NiMoO₄ surface. **g** Projected density of state (PDOS) of various exposed atoms in α -NiMoO₄-(110). **h, i** PDOS before (**h**) and after (**i**) H₂O being adsorbed on the Ni site of P- β -NiMoO₄.

10. Overall, there are a lot of papers investigating Ni-Mo species for HER. This study provides (when provided in more details) a protocol how to increase the surface area, enhanced even more with defective structure introduced by hypophosphite species. Performance is as good as other recent similar papers, and worth being reported, but my recommendation is to publish in electrochemistry specific journals because this study does not provide (or even mislead) new scientific content of the active site.

Response: We sincerely thank the reviewer's positive comments and the questions proposed, which do help to improve our paper's quality and depth.

(1) Indeed, NiMo-based materials are among the hottest electrocatalysts for HER. However, up to now, most efforts have been devoted to investigating the development of NiMo alloy, Ni-Mo sulfide, Ni-Mo nitride, and their composites. In contrast, little breakthrough was done on developing **cost-efficient non-noble transition metal oxide** as the HER electrocatalysts due to its inherent drawbacks.

(2) We put forward a novel non-solvent strategy to prepare achieve **phosphate substitution** and the subsequent **metastable beta crystal phase stabilization** of NiMoO₄, which could efficiently manipulate the **active electronic states** of NiMoO₄ and promote its intrinsic HER activity. Combined with detailed experiments and structural characterizations, phosphate substitution is proved to be imperative for the

stabilization of β -NiMoO₄, while oxygen vacancy plays a subordinate role in balancing the charge.

(3) P- β -NiMoO₄ exhibits a superior HER activity and stability at a large current density (1000 mA cm⁻²). The cost-efficient materials, the convenient non-solvent synthesis protocols, and the attractive comprehensive HER performance are practical and meaningful for potential industrial applications.

(4) We added an electrocatalytic performance comparison between our P- β -NiMoO₄ and the state-of-the-art NiMo-based electrocatalysts. We are confident our P- β -NiMoO₄ is comparable to or even better than these electrocatalysts.

Supplementary Table 5. The comparison of HER performance at 10 mA cm⁻² and 1000 mA cm⁻² for NiMo-based and CoMo-based electrocatalysts.

Electrocatalyst	Overpotential/mV	Overpotential/mV	Reference
P-NiMoHZ	23	210	This work
Ni ₃ N-NiMoN	31	/	Nano Energy . 44 , 353–363 (2018)
Ni-Mo/GC1h	95	/	ACS Catal. 10 , 12858–12866 (2020)
PS-MoNi@NF	30	/	Adv. Energy Mater. 11 , 203511 (2021)
Mo-NiO/Ni	50	/	ACS Energy Lett. 4 , 3002–3010 (2019)
MoS ₂ /FNS/FeNi	120	/	Adv. Mater. 30 , 1803151 (2018)
MoS ₂ /Ni ₃ S ₂	110	/	Angew. Chem. Int. Ed. 55 , 6702–6707 (2016)
Ni ₄ Mo	56	/	Angew. Chem. Int. Ed. 60 , 5771 – 5777 (2021)
N-NiCo ₂ S ₄	41	/	Nat. Commun. 9 , 1425 (2018)
NiMoO _x /NiMoS	38	236	Nat. Commun. 11 , 5462 (2020)
CoMoP@C	81	/	Energy Environ. Sci. 10 , 788 (2017)

NiMo ₃ S ₄	254	/	Angew. Chem. Int. Ed. 55 , 15240 (2016)
H-Fe-CoMoS	137	/	Nano Energy 75 , 104913 (2020)
Co ₂ Mo ₃ O ₈	38	/	Nano Energy 87 , 106217 (2021)

(5) In conclusion, we think this work not only use a novel strategy to prepare P-β-NiMoO₄, which shows outstanding HER activity and stability at small or large current density, but also study the science behind phase transformation and HER mechanisms. The introduction of active electronic states leads to a better understanding of the origin of HER activity, presenting a profound impact on exploring the low-cost non-noble transition metal oxide electrocatalysts. To this end, we sincerely show our great appreciation to reviewer for the professional and constructive comments. Following reviewer's suggestion, we are confident that the revised work is suitable for the scope of "Nature Communications".

Therefore, we sincerely expect this work to be published on this journal, which may be beneficial to more researchers. We sincerely hope that our carefully revised manuscript and added discussions could meet the requirements of the reviewer.

REVIEWER COMMENTS

Reviewer #1 (Remarks to the Author):

The revised manuscript is acceptable for publication.

Reviewer #3 (Remarks to the Author):

The authors improved the manuscript substantially and the argument regarding synthesis protocol and P doping becomes stronger. There remain two major concerns:

1. What is still missing is the surface information. All the characterization is dedicated to the bulk information. I am afraid that the authors did not understand the point about the literature for NiMo catalysts. The recent studies pointed out that even if you start with NiMo metallic alloy, the surface will reconstruct to make active surfaces for H₂O reduction. For this, there are multiple papers claiming that MoO_x species (oxidized state) is responsible, which may be similar structure at molecular levels to this current NiMoO₃ structure at the surface. It is well documented that the surface will be reconstructed. The catalyst is the surface phenomena: the HER with H₂O reduction (rather than H⁺ reduction) requires O-H bond dissociation of H₂O as the primarily rate limiting step. This type of discussion is totally lacking.
2. It is unclear how iR is compensated. Is the potential iR corrected? And if so, how the solution resistance is determined and how much was the resistance?

Point-by-point response letter

We thank the editor for giving us a chance to revise the manuscript, and also appreciate the reviewers for giving us constructive and high-quality suggestions, which would improve our work in depth. After careful evaluation of the manuscript and reviewers' comments, complementary experiments, characterizations and discussions are made. We hope the revised paper could meet the approval of both editor and reviewers.

The whole manuscript has been revised accordingly with changes clearly indicated by highlighting yellow. We have also paid special attention to the "Editor's comments" and make sure that our manuscript is consistent with the requirements of this journal. We ardently hope that in light of our responses to all the comments, the reviewers and readers can reach a more favorable view of the revised manuscript.

Please find point-to-point response to reviewers' comments attached below.

Reviewers' comments:

Reviewer #1 (Remarks to the Author):

The revised manuscript is acceptable for publication.

Response: We sincerely appreciate the reviewer again for his/her constructive and valuable comments in improving the quality of this manuscript.

Reviewer #3 (Remarks to the Author):

The authors improved the manuscript substantially and the argument regarding synthesis protocol and P doping becomes stronger. There remain two major concerns:

Response: Thanks for the reviewer's positive comments on our last version of the revised manuscript. Regarding to the rest two concerns, after carefully reading the reviewer's comments, we have added more experiments, characterizations, and discussions to support our arguments and strengthen the science behind our work.

1. What is still missing is the surface information. All the characterization is dedicated to the bulk information. I am afraid that the authors did not understand the point about the literature for NiMo catalysts. The recent studies pointed out that even if you start with NiMo metallic alloy, the surface will reconstruct to make active surfaces for H₂O reduction. For this, there are multiple papers claiming that MoO_x species (oxidized state) is responsible, which may be similar structure at molecular levels to this current NiMoO₃ structure at the surface. It is well documented that the surface will be reconstructed. The catalyst is the surface phenomena: the HER with H₂O reduction (rather than H⁺ reduction) requires O-H bond dissociation of H₂O as the primarily rate limiting step. This type of discussion is totally lacking.

Response: We really appreciate the reviewer's valuable and professional suggestion to make this work more convinced and precise. As mentioned by the reviewer, we also completely agree that surface reconstruction of electrocatalysts plays a significant role in electrocatalysis, which determines the actual active materials and sites. For NiMo alloy or its composite, surface reconstruction to give MoO_x species is a critical

phenomenon to enhance the HER performance of the electrocatalysts [Angew. Chem. Int. Ed. **60**, 7051–7055 (2021); ACS Catal. **10**, 12858-12866 (2020)]. Meanwhile, it is widely accepted that the surface reconstruction of NiMoO₄ under the oxygen evolution reaction condition (OER, positive potential vs. reversible hydrogen electrode) would lead to the in-situ formation of NiOOH, contributing to the improvement of the OER performance [Adv. Mater. **32**, 2001136 (2020); Adv. Energy Mater. 2101324 (2021)].

Actually, for P-NiMoHZ (phosphate substituted β -NiMoO₄) in our work, so far there were not many relevant discussions in this regard. Therefore, to unveil the surface state of phosphate substituted β -NiMoO₄ under HER condition, we have conducted several supplementary experiments and in-situ characterizations to reinforce the reliability of our work within one month.

(1) Firstly, we applied in-situ Raman spectroscopy to display the dynamic structure transformation of P-NiMoHZ at a series of potentials, ranging from 0 to -0.25 V vs. RHE.

Supplementary Figure 29. (a) The potential-dependent in-situ Raman spectra of P-NiMoHZ during HER process. (b) The ex-situ Raman spectra of P-NiMoHZ after HER electrocatalysis at a certain potential.

As shown in Supplementary Fig. 29a, the four spectra are almost the same, demonstrating that in the HER process the surface structure of the electrocatalyst remains unchanging. No new phase or new material comes into being, namely, no obvious surface reconstruction was detected. This result is consistent with the XRD patterns of the P-NiMoHZ before and after the stability test (Fig. 3e). By comparing

with the ex-situ Raman spectra of P-NiMoHZ (Supplementary Fig. 29b), in details, the peaks at 961.2 and 912.0 cm^{-1} are assigned to the symmetric and asymmetric stretching modes of the Mo=O bond, while the peak at 706.6 cm^{-1} belongs to the asymmetric stretching modes of Ni-Mo-O bonds [*Adv. Energy Mater.*, 2101324, (2021); *Ind. Eng. Chem. Res.* **46**, 2466-2472, (2007)].

Fig. 3 Electrocatalytic properties for HER. e XRD patterns of the P-NiMoHZ before and after the stability test.

(2) Besides the in-situ technique, we have additionally compared the pristine surface structure of P-NiMoHZ and the one after the long-term stability test by SEM and XPS. Obviously, the nanomorphology (Supplementary Fig. 32) and surface elemental valence states (Supplementary Fig. 30) still keep the same, further confirming the structure and material stability of P-NiMoHZ, as well as the surface stability.

Supplementary Figure 32. SEM images of the P-NiMoHZ before and after the stability test.

Supplementary Figure 30. (a) Ni 2p and (b) Mo 3d XPS spectra of the P-NiMoHZ before and after the stability test.

(3) Once confirming the surface structure stability, we further collected the electrolyte during the long-term stability test at 100 mA cm⁻² to clarify whether there was a dissolution of the active material on the electrode. As can be seen in Supplementary Fig. 31a, unlike the case of NiMoO₄ under OER condition (Figure R1), there is no distinct surface reconstruction platform and step in our potential-time curve. Through the analysis of the inductively coupled plasma-mass spectrometry (ICP-MS) (Supplementary Fig. 31b), it is clear that no material dissolution took place.

Supplementary Figure 31. (a) Long-term stability test of the P-NiMoHZ and (b) time-dependent concentration the corresponding electrolyte.

Figure R1. (a) Chronopotentiometric measurements of NiMoO₄ pre-catalyst and nickel foam (NF) at 10 mA cm⁻² in 1 M KOH with solution temperatures of T. (b) Schematic diagram for the temperature-dependent potential curve, which associates with the thermally induced reconstruction results. [*Adv. Mater.* **32**, 2001136 (2020).]

(4) According to the reviewer's suggestion, besides the added experiments, we have discussed more deeply on this issue.

Fig. 4 Calculated adsorption free energy and electronic structure of different adsorbates between α -NiMoO₄-(110) and P- β -NiMoO₄-(110) surface. **a** The optimized structure of P- β -NiMoO₄-(110). Especially, the Mo atom was substituted by the P atom according to the experiment verification. **b** The Gibbs free energy diagram on H₂O adsorbed on different sites in α -NiMoO₄-(110) and P- β -NiMoO₄-(110) surface, respectively. **c** Water dissociation barrier for reaction pathway of P- β -NiMoO₄ system.

The insets are the structure of the corresponding IS (initial state), TS (transition state) and FS (final state). **d** The corresponding charge density differences of H₂O adsorbed on Ni sites in P-β-NiMoO₄. The yellow and blue regions indicate the accumulated or dispersed amount of electron states of atoms around the interface, respectively.

To understand the primarily rate limiting step of the HER with H₂O reduction, we have presented an in-depth discussion regarding the relationship between hydrogen adsorption behavior and the water dissociation process, and their corresponding roles in the overall alkaline HER rate. For HER in alkaline conditions, there are two continuous steps of water dissociation and hydrogen adsorption. [*Angew. Chem. Int. Ed.* **57**, 7568-7579 (2018); *Adv. Sci.* **8**, 2001881 (2021); *Nat. Commun.* **12**, 3881 (2021)] For the water dissociation process, it may introduce an additional energy barrier and govern the overall reaction rate. The considerably slow rate of the water dissociation in alkaline electrolyte has greatly hindered the overall high purity hydrogen production and reaction kinetic rate. From the kinetic viewpoints, the activation barriers of water dissociation play an important role in accelerating to provide a neutral hydrogen source. As shown in **Fig. 4a-b**, it is revealed that the active Ni sites of P-β-NiMoO₄ system (Fig. 4a, b) exhibit much higher adsorption energy (-0.599 eV) for water adsorption, compared with α-NiMoO₄ system (-0.424 eV) (Supplementary Fig. 35). In addition, it is found that there is a linear correlation among H₂O adsorption energy, bond length and the amount of charge transfer Δe . The more active the electron transfer, the stronger the H₂O binding energy, and the shorter the α-NiMoO₄/P-β-NiMoO₄-H₂O bond length (Supplementary Fig. 36). Thus, the higher adsorption free energy of H₂O corresponds the lower activation barriers of water dissociation. In **Fig. 4c**, the active Ni sites of P-β-NiMoO₄ (b) system exhibit much lower activation barriers (0.569 eV) than that of α-NiMoO₄ (1.457 eV) and Pt catalysis (0.94 eV) (Supplementary Fig. 37) for water dissociation. [*J. Am. Chem. Soc.* **138**, 16174-16181 (2016); *Angew. Chem. Int. Ed.* **57**, 13163-13166 (2018)] Consequently, the P-β-NiMoO₄ system could accelerate water dissociation to provide neutral hydrogen source with the lower activation barrier for HER with H₂O reduction. From the microscopic view, the apparent alkaline HER activity is governed by two factors: the lower barrier to water dissociation and

appropriate (not too strong nor too weak) hydrogen adsorption. In Fig. 4d, the corresponding charge density differences of H₂O adsorbed on Ni site in P-β-NiMoO₄ are also represented, qualitatively reflecting the redistribution of electron states with the largest amount of charge transfer, further optimizing the decomposition of water. Thus, higher H₂O adsorption energy corresponds to the larger amount of charge transfer benefit from lowering the activation barrier of water dissociation.

(5) In conclusion, based on the above experiments and characterization, there is not obvious surface reconstruction in the HER process of P-NiMoHZ. We have carefully revised the manuscript and added discussions accordingly to strengthen our point (changes were marked by highlighted)

2. It is unclear how iR is compensated. Is the potential iR corrected? And if so, how the solution resistance is determined and how much was the resistance?

Response: Thanks for the reviewer's comments.

(1) iR is the ohmic potential drop caused by the flow of current in ionic electrolytes. Actually, the potential for driving the hydrogen evolution reaction (E_i) is composed of E_{HER} , iR, and the reaction overpotential η , as shown from the following equation:

$$E_i = E_{HER} + iR + \eta$$

Where E_{HER} is the Nernstian potential for the HER (equals to zero when referenced to a reversible hydrogen electrode).

According to the above formula, to display the overpotential more directly, we compensated iR at the level of 85% by the iR compensation program of the electrochemical workstation, which was also applied in other papers [*Materials Today Physics.*, **14**, 100253, (2020); *Energy Environ. Sci.*, **11**, 744-771, (2018)]. After setting the parameters, this program will measure the open circuit potential in the three-electrode system to determine the solution resistance before the linear sweeping voltammetry (LSV) test. Then, in the parameter setting stage of LSV, type in the resistance value to enable the iR compensation during the measurement.

(2) On the other hand, electrochemical impedance spectroscopy (EIS) is a more advanced technique for the determination of iR value [*Chem. Soc. Rev.*, **45**, 1529-1541,

(2016); *Materials Today Energy.*, **6**, 1-26, (2017)]. As shown in the Nyquist plots (Fig. 3c), the EIS spectrum of P-NiMoHZ sample (phosphate substituted β -NiMoO₄) displays a characteristic semicircle with a very small diameter which represent a low charge transfer resistance (R_{CT}). Based on the general descriptors in the Randles equivalent circuit (inset of Fig. 3c), R_s is the resistance of the solution. The value is $\sim 1.25 \Omega$.

Fig. 3 Electrocatalytic properties for HER. **c** Nyquist plots. The inset is the equivalent circuit schematic.

REVIEWERS' COMMENTS

Reviewer #3 (Remarks to the Author):

The authors sincerely addressed all the concerns raised by the reviewer. This oxide-based HER catalyst is very interesting and the paper is now recommended strongly for publication in Nature Communications.

Point-by-point response letter

We thank the editor for giving us a chance to revise the manuscript, and also appreciate the reviewers for giving us constructive and high-quality suggestions, which would improve our work in depth. After careful evaluation of the manuscript and reviewers' comments, complementary experiments, characterizations and discussions are made. We hope the revised paper could meet the approval of both editor and reviewers.

The whole manuscript has been revised accordingly with changes clearly indicated by highlighting yellow. We have also paid special attention to the "Editor's comments" and make sure that our manuscript is consistent with the requirements of this journal. We ardently hope that in light of our responses to all the comments, the reviewers and readers can reach a more favorable view of the revised manuscript.

Please find point-to-point response to reviewers' comments attached below.

Reviewers' comments:

Reviewer #3 (Remarks to the Author):

The authors sincerely addressed all the concerns raised by the reviewer. This oxide-based HER catalyst is very interesting and the paper is now recommended strongly for publication in Nature Communications.

Response: We sincerely appreciate the reviewer again for his/her constructive and valuable comments in improving the quality of this manuscript.